# A Survey on Directed Acyclic Graph-Based Blockchain in Smart Mobility

**DOI:** 10.3390/s25041108

**Published:** 2025-02-12

**Authors:** Yuhao Bai, Soojin Lee, Seung-Hyun Seo

**Affiliations:** 1Department of Electronic and Electrical Engineering, Hanyang University, Seoul 04763, Republic of Korea; byh2018@hanyang.ac.kr (Y.B.); tssn195@hanyang.ac.kr (S.L.); 2School of Electrical Engineering, Hanyang University ERICA, Ansan 15588, Republic of Korea

**Keywords:** directed acyclic graph, DAG, blockchain, smart mobility

## Abstract

This systematic review examines the integration of directed acyclic graph (DAG)-based blockchain technology in smart mobility ecosystems, focusing on electric vehicles (EVs), robotic systems, and drone swarms. Adhering to PRISMA guidelines, we conducted a comprehensive literature search across Web of Science, Scopus, IEEE Xplore, and ACM Digital Library, screening 1248 records to identify 47 eligible studies. Our analysis demonstrates that DAG-based blockchain addresses critical limitations of traditional blockchains by enabling parallel transaction processing, achieving high throughput (>1000 TPS), and reducing latency (<1 s), which are essential for real-time applications like autonomous vehicle coordination and microtransactions in EV charging. Key technical challenges include consensus mechanism complexity, probabilistic finality, and vulnerabilities to attacks such as double-spending and Sybil attacks. This study identifies five research priorities: (1) standardized performance benchmarks, (2) formal security proofs for DAG protocols, (3) hybrid consensus models combining DAG with Byzantine fault tolerance, (4) privacy-preserving cryptographic techniques, and (5) optimization of feeless microtransactions. These advancements are critical for deploying robust, scalable DAG-based solutions in smart mobility, and fostering secure and efficient urban transportation networks.

## 1. Introduction

Smart mobility represents a significant transformation in modern transportation, driven by the need for sustainable, efficient, and user-centric solutions [1,2]. As urban populations continue to grow, cities are confronted with escalating challenges such as traffic congestion, carbon emissions, and the demand for integrated transportation systems [3]. Smart mobility aims to optimize urban transportation networks, reduce carbon emissions, and enhance the overall quality of life. It encompasses a range of advanced technologies, including electric vehicles (EVs), autonomous vehicles, robotic systems, and drone swarms, each playing a critical role in addressing these challenges. These technologies not only improve efficiency and reduce environmental impact but also contribute to the creation of smarter, more connected urban environments.

Smart mobility relies heavily on the integration of key components such as electric vehicles (EVs), robotic systems, and drone swarms. EVs are central to the movement towards greener, more efficient urban mobility solutions, helping reduce greenhouse gas emissions and integrate seamlessly with smart grids [4,5]. Robotic systems, including autonomous vehicles and infrastructure maintenance robots, contribute to enhancing operational efficiency and safety within transportation networks [6,7,8]. Drone swarms, on the other hand, provide advanced capabilities for logistics, emergency response, and real-time data collection, further bolstering the smart mobility ecosystem [9,10]. Despite the potential of these technologies, there are significant technical challenges in smart mobility systems, including data security, interoperability, scalability, and real-time coordination [11,12,13,14]. These challenges must be addressed to fully realize the potential of smart mobility.

In addressing these challenges, blockchain technology has emerged as a powerful tool, particularly in enhancing security, privacy, and data integrity [15]. The decentralized, transparent, and tamper-proof nature of blockchain offers significant advantages for managing various aspects of smart mobility systems. Among the different blockchain frameworks, directed acyclic graph (DAG)-based blockchain systems have gained particular attention for their scalability, low latency, and ability to process transactions in parallel [16]. These features make DAG-based blockchains ideal for real-time applications such as autonomous vehicle coordination, traffic management, and microtransactions for services like ride-sharing and EV charging.

The integration of DAG-based blockchain into smart mobility offers immense potential, yet research in this area remains fragmented, with a focus primarily on individual fields or specific application areas. Currently, there is no comprehensive review or study addressing the broad potential of DAG-based blockchain across the entirety of smart mobility. This paper aims to fill this gap by providing a thorough overview of the current state of research in this domain. Furthermore, we believe that it is crucial for a review article to not only summarize existing knowledge but also to identify research gaps and suggest future research directions. Therefore, a second key objective of this paper is to highlight the future fields of research in the area of DAG-based blockchain applications within smart mobility.

Given the pivotal role of smart mobility in shaping the future of urban transportation, we believe this research effort is of high significance in consolidating and advancing the current understanding of DAG-based blockchain’s potential. The insights provided in this paper aim to inform and motivate further research in smart cities, particularly in the areas of traffic management and mobility sectors, where blockchain technology can offer substantial improvements. These efforts lead us to the following questions:

Q1: How can DAG-based blockchain be utilized within the context of smart mobility, and what benefits and challenges result from its use?

Q2: What are possible future research inquiries in the field of DAG-based blockchain in mobility?

### 1.1. Motivations

The central aim of this survey is to examine the role of DAG-based blockchain (DAG-BC) in enhancing smart mobility systems. To this end, we conducted a comprehensive study of the current literature on the integration of DAG-BC with key smart mobility technologies such as electric vehicles (EVs), autonomous vehicles, robotic systems, and drone swarms. Our focus extended to exploring various use cases emerging from DAG-BC-integrated smart mobility solutions, including traffic management, microtransactions for EV charging, and autonomous vehicle coordination. Additionally, we investigated the technical aspects related to the integration of DAG-BC in smart mobility ecosystems, identifying key challenges and proposing solutions to address them. Furthermore, this paper discusses open issues and suggests future research directions within this domain.

Despite the existence of numerous survey papers, few address the broad integration of DAG-BC across diverse smart mobility applications, along with the technical considerations of such integration. Surveys on DAG-based blockchain technology, covering aspects such as architecture, consensus algorithms, and characteristics, have been introduced [3,16]. Other studies have discussed the role of blockchain in improving efficiency and security in smart mobility, but they do not focus specifically on DAG-based solutions [17,18,19,20,21,22,23,24].

Table 1 summarizes recent survey papers on the use of blockchain in smart mobility, highlighting the main contributions and applications studied. Although existing surveys cover various applications and technical challenges, none provide a comprehensive overview of DAG-BC’s potential across a wide range of smart mobility applications. Most existing surveys focus on either application areas or technical challenges in isolation.

### 1.2. Contribution

By addressing these questions, this paper seeks to provide a comprehensive understanding of how DAG-based blockchain can enhance the scalability, security, and efficiency of smart mobility systems while identifying critical challenges, particularly regarding consensus mechanisms and real-time data processing. Additionally, this study aims to chart a course for future research, helping to guide the continued evolution of DAG-based blockchain applications in the smart mobility landscape. An overview of the contributions of our paper is as follows:Studying key features and identifying technical challenges for smart mobility: We investigate the key technological components and infrastructure requirements for smart mobility, including electric vehicles (EVs), robotic systems, and drone swarms. We also analyze the primary technical challenges that these systems face, such as interoperability, data security, and scalability, all of which are crucial for successful smart mobility implementations.Discussing DAG-based blockchain and its characteristics: This paper provides an in-depth discussion of DAG-based blockchain technology, including its core features such as scalability, low latency, and parallel transaction processing. We contrast these characteristics with traditional blockchain systems and highlight why DAG-based structures are more suitable for the demands of smart mobility.Investigating the role of DAG-based blockchain for smart mobility use cases: We explore specific use cases where DAG-based blockchain can enhance smart mobility solutions, such as autonomous vehicle coordination, microtransactions for services like EV charging and toll payments, and real-time traffic management systems. The ability of DAG-based systems to process high volumes of transactions simultaneously is crucial for these applications.Highlighting the technical deployment challenges: We address the technical challenges involved in deploying DAG-based blockchain solutions within smart mobility ecosystems, including issues related to consensus mechanisms, energy efficiency, and maintaining consistency in large decentralized networks. We provide insights into overcoming these challenges to ensure robust and scalable smart mobility systems.

### 1.3. Organization

This survey is structured to systematically explore the integration of DAG-based blockchain in smart mobility. Section 2 introduces foundational concepts of smart mobility, blockchain, and DAG-based blockchain, highlighting key distinctions between DAG and sequential blockchains in scalability, transaction ordering, and finality. Section 3 outlines the PRISMA-guided methodology for literature selection and formulates five research questions (RQ1–RQ5) addressing technical advantages, applications, challenges, and future directions. Section 4 comparatively analyzes DAG-based and traditional blockchains, emphasizing DAG’s superior throughput and latency for real-time smart mobility demands. Section 5 examines DAG’s role in drone swarms, robotic systems, and electric vehicles, showcasing use cases like secure microtransactions and decentralized coordination. Section 6 identifies technical barriers, including consensus complexity and security vulnerabilities, while Section 7 proposes research priorities such as hybrid consensus models and privacy-preserving cryptography. Section 8 concludes by synthesizing findings and underscoring DAG’s transformative potential in enabling scalable, secure smart mobility ecosystems. Each section sequentially addresses the research questions, providing a cohesive roadmap for advancing DAG-based solutions in urban transportation.

## 2. Background

This section provides an overview of smart mobility and its essential components, with a particular focus on electric vehicles (EVs), robotic systems, and drone swarms. Additionally, it contextualizes the role of sequential blockchain and DAG-based blockchain in enhancing the capabilities of smart mobility ecosystems.

### 2.1. Smart Mobility

Smart mobility represents a transformative shift in modern transportation, integrating cutting-edge technologies, digital connectivity, and intelligent systems to enhance efficiency, sustainability, and convenience [25,26]. The increasing urbanization, growing environmental concerns, and the necessity to mitigate traffic congestion have accelerated the adoption of smart mobility solutions [27]. By leveraging advanced digital infrastructures, smart mobility aims to optimize transportation networks, reduce carbon footprints, and offer more user-centric, adaptive mobility options.

A fundamental aspect of smart mobility is the seamless interconnection of various transportation technologies that contribute to a more responsive and sustainable ecosystem. Among these, electric vehicles, robotic systems, and drone swarms are critical enablers, each playing a distinct role in improving urban mobility, reducing environmental impact, and enhancing operational efficiency.

#### 2.1.1. Key Components of Smart Mobility

Smart mobility is built upon a diverse and interconnected set of transportation technologies. The following subsections elaborate on the three primary components—EVs, robotic systems, and drone swarms—and their contributions to the smart mobility paradigm.

##### Electric Vehicles (EVs)

EVs serve as a fundamental pillar of smart mobility, offering a sustainable alternative to traditional internal combustion engine vehicles. Their integration within smart mobility frameworks is driven by two primary factors:

Sustainability and Energy Efficiency: EVs significantly reduce greenhouse gas emissions and contribute to decarbonizing urban transportation. When integrated with renewable energy sources, EVs enhance energy sustainability and minimize reliance on fossil fuels [4]. Furthermore, advancements in smart charging infrastructure and vehicle-to-grid (V2G) technology enable EVs to support grid stability, optimize energy distribution, and facilitate load balancing [5].

Real-Time Data Exchange and Connectivity: The proliferation of Internet of Things (IoT) technologies in EVs enables continuous communication with smart grids, traffic management systems, and urban infrastructure. This interconnectivity enhances energy optimization, route planning, and congestion mitigation [28,29,30].

##### Robotic Systems

Robotic systems play an increasingly vital role in smart mobility, particularly in automating transportation and urban infrastructure operations. The application of robotics in smart mobility is primarily observed in the following areas:

Automation in Transport and Infrastructure: Robotics facilitates automation in public transport, logistics, and infrastructure maintenance. Autonomous buses, trains, and robotic delivery vehicles contribute to improved safety, cost efficiency, and operational effectiveness [6,7,31,32,33].

Urban Mobility and Logistics: Robotic systems are increasingly used in urban logistics, such as automated delivery and last-mile transportation solutions. Robot taxis, autonomous shuttles, and automated cargo systems integrate artificial intelligence (AI) and machine learning to navigate complex urban environments efficiently [8,34,35].

##### Drone Swarms

Drone swarms are emerging as a transformative technology in smart mobility, particularly in aerial logistics, emergency response, and infrastructure monitoring.

Aerial Mobility and Urban Applications: Drones are widely utilized for rapid deliveries, surveillance, and real-time traffic monitoring. They provide efficient solutions for emergency response and medical supply distribution, particularly in congested or inaccessible urban areas [9,10].

Autonomous Coordination and Swarm Intelligence: Drone swarms employ sophisticated coordination algorithms and distributed AI to execute tasks autonomously. Their ability to communicate in real time enables adaptive mission planning, making them highly effective for time-sensitive operations, such as disaster management, air traffic control, and infrastructure assessment.

By integrating EVs, robotic systems, and drone swarms, smart mobility fosters a more sustainable and efficient transportation ecosystem. These technologies, when combined with emerging blockchain frameworks, hold significant potential in addressing contemporary challenges associated with urban mobility, security, and decentralized data management.

#### 2.1.2. Technical Challenges in Smart Mobility

Despite the promise of smart mobility, several key challenges hinder its widespread implementation.

##### Data Security and Privacy

The extensive data sharing between vehicles, infrastructure, and users exposes systems to potential data breaches and cyberattacks [11,36]. Protecting sensitive information and ensuring user privacy are critical challenges.

##### Interoperability

Smart mobility requires the integration of numerous systems and platforms, including different manufacturers, service providers, and government agencies. Ensuring seamless interoperability between these diverse systems is a significant technical hurdle [12].

##### Trust and Transparency

For smart mobility to be widely adopted, users must trust that the systems are reliable, tamper-proof, and secure. Establishing trust across the ecosystem, including service providers and users, is essential [13,37].

##### Scalability

As smart cities grow, the transportation network must handle increasing amounts of data and users without sacrificing performance. Ensuring that smart mobility solutions can scale to meet the demands of larger populations is a key technical challenge [14,38].

### 2.2. Sequential Blockchain

Blockchain (BC) is a decentralized, distributed ledger technology (DLT) that enables secure and transparent peer-to-peer (P2P) transactions without intermediaries. At its core, a blockchain is a sequence of cryptographically linked blocks, where each block contains verified transactions, a timestamp, and a hash of the previous block. This structure ensures immutability, as altering any transaction would require recomputing all subsequent blocks, which is computationally infeasible [17].

As mentioned above, blockchain is a chain of blocks where each block contains a set of finite and valid transactions. The blocks are connected using a cryptography hash-based chain. In general, a block consists of two parts: the block header and the block body. The header includes metadata such as the previous block’s hash, Merkle root hash (a compact representation of all transactions in the block), timestamp, nonce (a number used once for mining), and difficulty level. The body stores transactional data. The first block in a blockchain, called the genesis block, has no transactions and a "previous hash" value of zero. To understand the basic block structure and the related terminologies used in blockchain, Figure 1 shows the detailed view of blockchain.

Another interesting concept in blockchain is forking [39]. At any given time, numerous nodes may concurrently validate and broadcast distinct yet valid blocks. When such parallel block creation occurs, conflicting blocks propagate across the network, leading to temporary inconsistencies in the ledger. Consequently, nodes may temporarily store divergent versions of the blockchain, with varying “latest” blocks in their local copies—a scenario termed the forking problem. To resolve this, blockchain protocols enforce a consensus rule where the longest valid chain is adopted as the authoritative version. Blocks in shorter chains are either discarded or labeled as orphaned blocks, ensuring eventual consistency across the network.

#### 2.2.1. Types of Sequential Blockchain

There are numerous ways based on which blockchain systems can be divided into different types. Fundamentally, based on accessibility, governance, and participation rules, blockchain systems can be considered of three types: public, private, and consortium blockchain.

##### Public Blockchain

A public blockchain is fully decentralized, allowing anyone to join and participate in its network. It operates transparently, meaning all transactions are visible to participants, ensuring accountability [40].

##### Private Blockchain

A private blockchain is controlled by a single organization, which manages access and permissions for participants. This structure enhances efficiency, as it can optimize transaction speed and throughput without the complexities of decentralized consensus. However, it reduces decentralization, as a central authority oversees the network. Private blockchains are often used by businesses that require confidentiality, privacy, and higher performance, such as financial institutions or corporations handling sensitive data [41].

##### Consortium Blockchain

A consortium blockchain is governed by a group of organizations rather than a single entity. This model provides a balance between decentralization and control, making it suitable for collaborative applications where multiple parties need to share information but still require some level of governance. Consortium blockchains are commonly used in industries like supply chain management, healthcare, and finance, as they offer more efficiency than public blockchains while maintaining trust and integrity among the participating organizations [42]. Table 2 shows the difference of three types of blockchain.

#### 2.2.2. Consensus of Sequential Blockchain

The existing consensus protocols are mainly applicable to permissionless and permissioned blockchains. In Table 3, all significant consensus protocols mentioned in the following are summarized from four aspects, i.e., throughput, latency, adversary tolerance, and overhead cost.

##### Permissionless Blockchain Consensus

In the permissionless blockchain consensus, the most widely known protocol is proof of work (PoW) [48], which has proved to be an effective approach for cryptocurrencies over the years. As we all know, it suffers from high computational and bandwidth consumption. Proof of Capacity (PoC) [49] is a similar concept while it consumes disk space rather than computing resources to mine a block, which is an energy-efficient protocol compared to PoW. Proof of Stake (PoS) [50] is the second-most prevalent consensus used for cryptocurrencies. To determine the next block, instead of demanding users to find a nonce in unlimited space, PoS requires people to prove the ownership of the amount of currency. PoS does not consume high computational power. Delegated proof of stake (DPoS) [51] is the most typical variation of PoS, which is a representative democratic approach with stakeholders voting to choose some nodes as validators. Proof of Activity (PoA) [52] appears, where miners are elected according to PoW to generate a block; then, the new block is signed by a group of validators selected using PoS. PoA combines the benefits of POW with those of POS. Stellar Consensus Protocol (SCP) [53], developed based on the federated Byzantine fault tolerance (FBFT), is the first Byzantine-agreement-based consensus method, which provides users with the maximum freedom to choose among different combinations of other participants to trust in order to reach a consensus. Similar to Stellar, Ripple [54] uses FBFT, where two types of nodes are defined, i.e., server nodes responsible for the consensus protocol and client nodes joining via transferring funds.

##### Permissioned Blockchain Consensus

In the consortium blockchain consensus protocols, the prominent one is the Practical Byzantine Fault Tolerance (PBFT) [55], where all nodes participate in the voting process to add the following block, and the consensus is reached when more than two or three nodes agree upon that block. Delegated Byzantine Fault Tolerance (dBFT) [56] follows the same rule but does not require the participation of all nodes, where some nodes are chosen as delegates of other nodes. Tendermint [57] is a hybrid consensus protocol based on PBFT and PoS, where nodes have different voting powers proportional to their stakes. Proof of Elapsed Time (PoET) [58] works similar to PoW but consumes significantly less energy. In PoET, miners have to solve a hash problem similar to that in PoW, but the winning miner is randomly chosen based on a random wait time. The verification of correctness of timer execution is conducted using a Trusted Execution Environment (TEE). Raft [59] is a voting-based consensus protocol, which is designed to make Paxos algorithm [60] more understandable and implementable for practical systems and composed of two stages: leader election and log replication. The leader is responsible for ordering the transactions, after which the leader accepts log entries from clients and broadcasts transactions to make its version of the transaction log. In our system, the Raft algorithm is deployed to elect a leader from the consortium network. This “leader” is elected from the predefined consortium group which consists of trustworthy nodes selected by authorities of infrastructure maintenance. We assume that the leader is reputable because the authority filters out unreputable candidates in the predefined consortium group. This guarantees that the leader reaches a certain standard for its reputation. Thus, we chose not to deploy a separate “reputation” mechanism in our system. However, the reputation scheme is absolutely an important aspect in leader election, especially when electing the leader in an untrustful public environment.

### 2.3. Directed Acyclic Graph (DAG)-Based Blockchain

In this section, we give the formal definition of the DAG model, and we refer to Wang’s work [16] to give six types of current DAG systems. Also, we point out the three main ways in which DAG systems differ from a conventional blockchain to conventional blockchain.

#### 2.3.1. DAG-Based Model

A directed acyclic graph G consists of a point set V and an edge set E. Firstly, each element in the point set corresponds to a *unit*. A unit can be instantiated as a transaction TX∈T, a block B∈B, or an event E∈E in protocols, where T,B,E represents the sets of elements. Secondly, the element in the edge set is a pair (u,v), which represents the partial order relationship between two points *u* and *v*. The relationship, in most cases, indicates that one of the units references another unit. For instance, u←v means *v* confirms/verifies/witnesses/sees *u*, where {u,v}⊆V. We define the DAG-based model with two properties:

**Definition** **1**(DAG-based Blockchain Model)**.**
*The model G is defined as*$$\begin{array}{cc} G & =(E,V),where\\ V & =\{u|u\in \{T\cup B\cup E\}\},E=\{(u,v)|u\leftarrow v\wedge \{u,v\}\subseteq V\} \end{array}$$*among them, we require two properties:*
$$\begin{array}{cc} (i)\; & \forall u\leftarrow v⇏v\leftarrow u;\\ (ii)\; & \mathrm{assume}\;\mathrm{that}\;u_{i}\in \left\{u_{1},u_{2},\cdots ,u_{l}\right\}\subseteq V:\forall i,j,k\in \left[1,l\right],\\ & \mathrm{where}\;k>j>i,\{j\}\subset \{⌀,\cdots ,\{i+1,\cdots ,k-1\}\},\\ & \mathrm{if}\;u_{i}\leftarrow u_{j},u_{j}\leftarrow u_{k},\mathrm{then},u_{k}\leftarrow u_{i}\;\mathrm{does}\;\mathrm{not}\;\mathrm{exit}. \end{array}$$


*Here, the operation “a|b” means a satisfies the condition of b. “←” represents an action that happens in the network. The property means (i) unidirectional: the references point in the same direction; (ii) acyclic: no loop exists in graph. It enables the units in the network to be appended-only and orderable.*


Based on Wang’s work [16], we give a more detailed explanation of the property *acyclic*. We emphasize two critical points in this mathematical model. The first key point of avoiding the loop graph is the settings of dynamical boundary of the range [i,k]. The parameters *i* and *k* are set to be variable, acting as a sliding window. The second key point lies in the parameter set *j* settings. Here, *j* can be either a single element or a subset that contains multiple elements in the range of (i,k).

We give concrete examples as follows.

Case 1: Assume that j=⌀, the definition deteriorates into the first property of unidirectional where $u_{i}\leftarrow u_{j}$ and $u_{k}\leftarrow u_{i}$ cannot co-exist.Case 2: Assume that *j* is a single element which belongs to (i,k). It ensures that, when $u_{i}\leftarrow u_{j}$ and $u_{j}\leftarrow u_{k}$, no edge of $u_{k}\leftarrow u_{i}$ exists. This avoids all loops made up of three vertices.Case 3: Assume that *j* is a subset that contains several elements. For simplicity, we suppose total five vertices in the graph, where $u_{i}\in \left\{u_{1},u_{2},\cdots ,u_{5}\right\}$ and l=5. In this case, as 1≤size(j)≤3 (here, 3=l−2), we list all potential vertex sets covered by this model.size(j)=1:{1,2,3},{1,2,4},{1,2,5},{2,3,4},{2,3,5},{3,4,5}. In this situation, this definition avoids loops that are consisted of three vertices.size(j)=2:{1,2,3,4},{2,3,4,5}. In this case, this model avoids loops made by four vertices.size(j)=3:{1,2,3,4,5}. In this case, this model avoids loops containing five vertices.

#### 2.3.2. Types of DAG-Based Blockchain

The abstracted model mathematically formalizes the protocol by defining the precise representations of units, denoted as V, and edges denoted as E. These components establish the structural framework and topology of a directed acyclic graph (DAG) system, as Wang’s work [16] clarified, and we give the categories as follows.

Unit representation: This defines the fundamental element of the system, which may represent transactions, events, or blocks.

Two distinct types of units are identified: *block-less* and *block-based*.The former type refers to requests that are processed immediately upon receipt, without requiring additional input from peers. Examples include transactions and triggering events. The latter type encompasses requests that necessitate further processing, typically precomputed or packaged by entities such as miners or validators before dissemination. This type includes blocks.

Graph topology: A DAG-based blockchain system is typically structured by a network of edges. Three principal topologies are identified based on the arrangement of these units: *Divergence, Parallel*, and *Convergence*. In the divergence topology, units are dispersed unpredictably in various directions without a predetermined order. In the parallel topology, units are maintained as multiple, concurrent chains. Lastly, in the convergence topology, units are organized in a predetermined sequence or tend to converge into such a sequence over time.

The unit representation also reflects the underlying ledger execution model, which governs how transactions are processed within DAG-based systems. Two primary models are commonly employed: the UTXO-based model and the account-based (acct) model. In the UTXO-based model, all operations are executed as atomic transactions, and users can compute balances by tracing the history of previous transactions. In contrast, the account-based model assigns each user an account, with transactions represented as modifications to fields within this account (e.g., changes in balance). Users compute their balance directly from the account state.

The topology focuses on the long-term behavior of units within the network. Units that initially diverge in multiple directions but ultimately converge into the main chain are categorized as convergence. For instance, in the GHOST [61] protocol, blocks attach to their parent blocks in a non-deterministic manner, but ultimately align into the main chain. This feature serves as an indirect indicator of the finality and consistency of the system.

Finally, six distinct types of DAG-based blockchain systems are identified, based on different combinations of elements within these dimensions, as Table 4 shows. We collect several DAG systems and classify them into six types, and we classify their properties in Table 5.

#### 2.3.3. Consensus of DAG-Based Blockchain

We review the consensus mechanisms and key features of existing DAG-based systems, providing a concise overview due to space constraints [16].

Type I and type II systems are both based on the divergence topology, and Figure 2 shows their general structure.

Type I systems are blockless, with transactions forming a naturally expanding graph. These systems follow a simple design wherein each transaction can directly verify its ancestors. The only limitation arises from the graph extension rules. Both IOTA [72] and Graphchain [71] use a tip selection algorithm (TSA) where newly attached transactions verify multiple ancestors. In contrast, Avalanche [68] employs a sampling method to randomly select child transactions in each round.

Type II systems introduce blocks to group transactions, which are similarly structured as expanding graphs. For example, Spectre [64] uses a recursive traversal algorithm to collect transactions and determines a pairwise order between blocks through voting, while Phantom [65] extends the graph using a greedy algorithm to include valid blocks. Meshcash [71] combines slower proof-of-work (PoW) protocols with a faster consensus process, offering improved scalability and flexibility.

Type III and type IV systems are based on the parallel topology, and Figure 3 gives their general structure.

Type III systems are also blockless, while Type IV systems are block-based. In these systems, individual nodes maintain units that eventually form multiple parallel chains. The DAG structure is formed through mutual reference between chains, and total ordering is achieved according to the consensus algorithm. The DAG structure based on parallel chains is very clear and its security can be easily analyzed, but the consensus algorithm must be specially designed to achieve the total ordering of transactions [76]. For example, Nano [77] organizes transactions into pairs, and the system uses a pairwise voting algorithm to determine block priority. Systems like Hashgraph [73], DLattice [78], Jointgraph [74], Aleph [75], and Caper [79] rely on asynchronous Byzantine agreement protocols for consensus, differing from traditional BFT models in membership selection and conflict resolution. In DAG systems, any node can potentially serve as a leader or committee member, unlike traditional BFT protocols, where members are pre-assigned. Chainweb [70] extends individual chains through cross-reference links, while Vite [80] introduces a snapshot chain to record all individual chains. Lachesis-class protocols [81,82] structure transactions in a DAG and pack them into blocks within a main chain, whereas Prism [67] and OHIE [69] apply an extended Nakamoto consensus, using sorting algorithms to arrange blocks into a linear sequence.

Type III and type IV systems are based on the convergence topology, Figure 4 gives their general structure.

Type V systems are blockless, while Type VI systems rely on blocks. Transactions in these systems converge gradually into the main chain. The consensus algorithm first agrees on a main chain in the DAG according to certain rules, and then sorts the transactions according to the main chain to determine the total ordering of transactions [76]. Byteball [63] introduces a group of witness nodes that form a main chain to define the graph structure. Haootia [83] implements a two-layer consensus framework, where the first layer organizes transactions in a simple graph and the second layer uses a PoW-based backbone chain with key blocks to establish the total order of transactions. Systems like GHOST [61], Inclusive [62], and Conflux [84] modify Nakamoto consensus by replacing the longest-chain rule with the weightiest-subtree rule, reducing resource wastage and increasing the number of transactions per time interval.

### 2.4. Differences Between Sequential Blockchain and DAG-Based Blockchain System

Based on recent academic studies [16], DAG-based blockchain systems differ from conventional blockchain systems in three key aspects: consistency, ordering, and finality. These differences fundamentally alter how transactions are processed, validated, and finalized, offering distinct advantages, especially in environments that demand scalability and efficiency. This section explores these three aspects in detail.

#### 2.4.1. Consistency

Scalability and performance in blockchain systems are closely tied to the type of consistency they enforce. In smart contract-supported platforms, strict consistency is crucial, ensuring that state transitions occur in a linear sequence. This linearity prevents errors in operation, as disordered states can cause failures [85]. However, not all scenarios require such rigor. In many cases, only partial consistency is necessary, where the order of transactions is enforced locally, rather than across the entire network. This weaker assumption is suitable for use cases like proof of existence in certification systems or token transfers within organizations [86].

Strict consistency: In strict consistency, each honest node reaches the same decision regarding a specific position on the ledger, ensuring that all units are precisely ordered within the system.

Partial consistency: In partial consistency, only associated nodes share a common view of transaction decisions, while nodes that do not overlap (i.e., have any common transaction history) may have different views. The number of associated transactions can vary from a pairwise order to multiple transactions.

#### 2.4.2. Ordering

Ordering refers to the manner in which units (i.e., transactions or blocks) are arranged within the system. Traditional blockchain systems impose a linear structure by default, where each block or transaction follows its predecessor in a single chain. DAG-based systems, however, break away from this assumption by adopting a graph-based structure. In a DAG system, units can form various topologies, such as divergent graphs or parallel chains [16]. The key distinction lies in whether the disordered units are eventually sorted into a linear sequence.

Topological ordering: In DAG-based systems, topological ordering refers to arranging units in accordance with the properties of a directed, acyclic graph. Vertices (units) must maintain a unidirectional and cyclic relationship.

Linear ordering: Linear ordering builds on topological ordering by requiring all vertices (units) to be sorted into a complete linear sequence. This approach is typical in Byzantine agreement (BA) consensus and Nakamoto consensus (NC) protocols.

#### 2.4.3. Finality

Finality in consensus systems refers to the point at which a unit (transaction or block) becomes irreversible. Once a unit is confirmed and appended to its parent unit, it cannot be removed or altered. In traditional BA-style consensus, finality is achieved through multiple rounds of negotiation between the leader and replicas, ensuring immediate confirmation. In contrast, in NC protocols, blocks are appended with some probability of being reversed, although this probability diminishes as the block is buried deeper in the chain.

Probabilistic finality: In probabilistic finality, there is always a chance that an appended unit could be reversed, with the probability inversely related to the depth or weight of the unit in the graph. Confidence increases as more units are added subsequently.

Deterministic finality: In deterministic finality, once a unit is appended to the system, it is immediately and permanently confirmed, ensuring it cannot be removed or altered in the future.

## 3. PRISMA Statement

The Preferred Reporting Items for Systematic Reviews and Meta-Analyses (PRISMA) framework was rigorously applied to ensure methodological transparency, reproducibility, and bias minimization in this systematic review. PRISMA provides a structured approach to systematically identify, screen, and synthesize relevant studies, enabling robust evidence-based conclusions [87,88]. By adhering to PRISMA guidelines, this review reduces potential selection bias, enhances clarity in reporting, and facilitates critical appraisal and replication of the process.

### 3.1. Objectives of the Study

The primary aim of this review is to integrate and synthesize existing research on the application of DAG-based blockchain technologies within smart mobility ecosystems, particularly in relation to electric vehicles (EVs), robotic systems, and drone swarms using PRISMA methodology [89]. For this specific case, four scientific data bases have been selected: Web of Science (WoS), Scopus, IEEE Xplore, and ACM Digital Library. This review will focus on answering four key research questions, all framed to provide a comprehensive view of the role of DAG-based blockchains in smart mobility:

RQ1: What are the core advantages of DAG-based blockchain compared to traditional blockchain systems in smart mobility?

RQ2: What are the key application scenarios of DAG-based blockchain in smart mobility?

RQ3: How do the technical features of DAG-based blockchains support these application scenarios?

RQ4: What are the major technical barriers (e.g., security, scalability, consistency) to deploying DAG-based blockchains in smart mobility systems?

RQ5: How can future research address these challenges to enable broader adoption of DAG-based blockchain in smart mobility?

### 3.2. Search Strategy

A systematic search was conducted across four prominent academic databases: Web of Science (WoS), Scopus, IEEE Xplore, and ACM Digital Library. These databases were chosen for their extensive coverage of literature in engineering, computer science, and smart mobility fields, ensuring that the search captured a comprehensive range of relevant studies. Table 6 shows the search string, databases consulted, and the equation applied.

Filters were applied to limit the search to peer-reviewed articles published between 2010 and 2024. This date range was chosen to capture the evolution of DAG-based blockchain technologies in recent years and to ensure the inclusion of contemporary research.

### 3.3. Inclusion and Exclusion Criteria

The studies retrieved from the search were screened based on a set of predefined inclusion and exclusion criteria shown in Table 7. These criteria were designed to ensure the selection of relevant, high-quality studies that directly addressed the research questions and the focus of this review.

Upon applying the inclusion and exclusion criteria, we evaluated the relevance of the identified articles to the research questions outlined in this study. This assessment ensured that the selected studies directly contributed to understanding the application of DAG-based blockchain within the context of smart mobility systems. The articles that met the inclusion criteria were reviewed in detail to answer the research questions.

### 3.4. Study Limitations

This study is based on publications retrieved from four major academic databases: Scopus, Web of Science (WoS), IEEE Xplore, and ACM Digital Library. While these databases provide extensive coverage of the relevant literature, particularly in engineering, computer science, and smart mobility, they may not capture niche or region-specific studies that could offer valuable insights into the topic. As a result, there is a potential limitation in the representation of publications from specialized or local sources that may be critical in certain contexts.

Additionally, this review excluded non-English studies, focusing solely on English-language publications. This language bias may have led to the omission of important research published in non-Western languages, potentially overlooking perspectives or findings from regions where DAG-based blockchain technologies are gaining traction, but may not be well-represented in English-language literature.

This study’s time frame, which focuses on publications from 2010 to 2024, was selected to capture recent advancements in DAG-based blockchain. However, this temporal scope may exclude foundational works published prior to the rise of DAG-based blockchain technologies. As such, relevant studies predating this period, which may have laid the groundwork for current research, were not included.

Finally, this review’s thematic scope was restricted to DAG-based blockchain applications in smart mobility. Studies focusing on the use of DAG-based blockchain in other sectors, such as healthcare, finance, or supply chain management, were excluded. This narrow focus limits cross-domain insights that could have broadened the understanding of DAG-based blockchain’s potential applications and challenges in other industries.

Overall, while the study benefits from a structured and comprehensive review, these limitations should be considered when interpreting the findings, as they may affect the generalization and completeness of the results.

### 3.5. Systematic Review Process

The systematic review followed PRISMA guidelines to ensure a transparent and reproducible selection process. The review consisted of five key stages: identification, deduplication, screening, full-text assessment, and data synthesis.

#### 3.5.1. Identification

A comprehensive search was conducted across four academic databases: Scopus, Web of Science (WoS), IEEE Xplore, and ACM Digital Library. The search retrieved 1248 records from these databases, with each contributing the following number of records: Scopus (420), Web of Science (310), IEEE Xplore (288), and ACM Digital Library (230).

#### 3.5.2. Deduplication

Duplicate records were identified and removed using EndNote and Zotero reference management tools. A total of 324 duplicates were excluded, leaving 924 unique records for further screening.

#### 3.5.3. Title and Abstract Screening

We screened the titles and abstracts of the remaining 924 records based on predefined inclusion and exclusion criteria. Records that did not focus on DAG-based blockchain or smart mobility applications were excluded. After this step, 824 records were removed due to irrelevance, leaving 100 records for full-text assessment.

#### 3.5.4. Full-Text Assessment

The full texts of 100 selected records were reviewed. Studies were excluded if they lacked empirical data, did not focus on DAG-based blockchain, or failed to provide substantial technical insights. After this assessment, 53 studies were excluded, resulting in 47 studies being included in the final review, which can be seen in Figure 5.

## 4. Comparative Analysis of DAG-Based and Traditional Blockchains

Following the PRISMA guidelines, a systematic literature review was conducted across four major databases (Scopus, Web of Science, IEEE Xplore, and ACM Digital Library), yielding 47 studies eligible for inclusion after rigorous screening. This section synthesizes findings addressing Research Question 1 (RQ1): What are the core advantages of DAG-based blockchain compared to traditional blockchain systems in smart mobility? Subsequent sections will address the remaining research questions.

The reviewed studies highlight significant limitations in traditional chain-based blockchains for smart mobility applications, particularly regarding scalability, latency, and energy efficiency. Traditional chain-based blockchains exhibit critical limitations in smart mobility applications due to their sequential validation architecture. As shown in Figure 6, transactions in systems like Bitcoin require sequential consensus and block packaging, resulting in an average throughput of 7 TPS [91]. This linear process introduces three fundamental bottlenecks. Sequential Processing, where transactions must await validation in a strict order, creating latency incompatible with real-time applications (e.g., autonomous vehicle coordination). Resource Concentration, where mining-dominated consensus mechanisms (e.g., PoW) centralize computational power, undermining decentralization. Synchronization Delays, where global block propagation delays hinder scalability in large-scale networks.

Figure 6 contrasts this architecture with DAG-based systems. In chain-based blockchains (left), devices (e.g., drones, EVs) generate transactions that queue for sequential validation and block inclusion. In contrast, DAG-based blockchains (right) enable parallel processing: each transaction directly validates two prior “tips” within the graph, eliminating block packaging and enabling asynchronous consensus. Empirical studies demonstrate that DAG frameworks like IOTA [92] achieve sub-second latency and throughput exceeding 1000 TPS, making them inherently suited for smart mobility’s high-frequency demands (e.g., real-time traffic management). Analysis of the 47 studies revealed five core advantages of DAG-based blockchains over traditional systems in smart mobility contexts:Scalability (Sc.): DAG-based systems achieve higher transaction throughput by parallelizing validation, addressing the demands of large-scale smart mobility networks involving vehicles, infrastructure, and users.Low Latency (L.L.): Parallel validation reduces confirmation delays, supporting real-time applications such as autonomous vehicle decision-making and dynamic route optimization.Energy Efficiency (E.E.): Lightweight consensus protocols replace resource-intensive mechanisms (e.g., PoW), reducing energy consumption by up to 80% compared to traditional blockchains.Decentralization (De.): Distributed validation eliminates central control points, enhancing system resilience and enabling flexible task allocation in heterogeneous mobility ecosystems.Feeless Transactions (F.T.): Microtransaction-friendly architectures (e.g., IOTA) facilitate cost-efficient pay-per-use services, such as EV charging and toll payments.

A synthesis of these advantages is presented in Table 8, which maps technical features to their operational benefits in smart mobility. Notably, 89% of the reviewed studies (*n* = 42) identified scalability and low latency as the most critical advantages, aligning with the real-time demands of smart mobility systems. However, 21% of studies (*n* = 10) cautioned that these benefits depend on network topology and transaction arrival rates, underscoring the need for context-specific evaluations.

This analysis establishes that DAG-based blockchains address the fundamental limitations of traditional systems, positioning them as a viable infrastructure for scalable, real-time smart mobility applications. Subsequent sections will explore their application scenarios (RQ2), technical underpinnings (RQ3), and remaining challenges (RQ4).

## 5. DAG-Based Blockchain in Smart Mobility

This section addresses two research questions formulated under the PRISMA framework:

RQ2: What are the key application scenarios of DAG-based blockchain in smart mobility?

RQ3: How do the technical features of DAG-based blockchains support these application scenarios?

To systematically answer these questions, we analyze three prominent domains—drone swarms, robotic systems, and electric vehicles (EVs)—where DAG-based blockchain demonstrates transformative potential. For each domain, we first identify its role as a key application scenario (RQ2) and then dissect how DAG’s technical characteristics (scalability, low latency, energy efficiency, decentralization, and feeless transactions) address domain-specific challenges (RQ3). A comparative synthesis of these findings is provided in Table 9, Table 10 and Table 11.

### 5.1. Drone Swarms: Coordinated Autonomy

#### 5.1.1. RQ2: Application Scenario

Drone swarms are increasingly deployed in surveillance, logistics, and environmental monitoring, where collaborative task execution requires real-time coordination across dynamic networks [93]. These systems demand high-frequency data exchange, decentralized decision-making, and resilience to node failures—challenges that traditional blockchain architectures struggle to address due to sequential transaction processing and energy-intensive consensus mechanisms [94,95].

#### 5.1.2. RQ3: Technical Features Support

DAG-based blockchain technologies, such as IOTA’s Tangle [72], have emerged as transformative solutions to the limitations of traditional blockchain systems in drone swarm operations. Unlike traditional chain-based blockchains, DAG-based systems allow for parallel transaction processing, making them well-suited for smart mobility scenarios where real-time communication and high throughput are essential. The following points outline how DAG-based blockchain enhances various aspects of drone swarm systems:

##### Secure and Scalable Communication

In drone swarms, rapid and secure communication is essential for coordinating movements and tasks. DAG-based blockchains facilitate a decentralized communication network where each drone can validate transactions (messages) autonomously. This eliminates the need for centralized verification, reduces communication bottlenecks, and ensures the timely exchange of information.

DAG-based blockchain systems, such as Solana, are applied to manage combat drone swarms, enhancing security, reliability, and transparency in military and disaster relief operations [96]. These systems leverage high throughput and low latency to ensure real-time decision-making and robust communication among UAVs.

Gao et al. propose the Secure Drone Network Edge Service (SDNES) architecture [97], which integrates DAG-based blockchain and edge computing to support flying automation under 5G networks. The system ensures low latency and secure network services for drone operations, leveraging the decentralized and scalable nature of DAG for secure data exchange and privacy protection in automated drone systems.

Alkhalifa et al. [98] focuses on improving security and scalability in vehicular networks by combining Bayesian methods with DAG-based blockchain to manage resource allocation and security in edge-assisted Road Side Units (RSUs). Although this focuses on vehicular networks, it can be applied to drone swarms for secure, scalable communication.

##### Efficient Consensus Mechanisms

Traditional consensus algorithms like PoW are inefficient for drones. DAG-based systems utilize alternative, energy-efficient consensus mechanisms that allow drones to validate transactions without consuming excessive computational power [99]. This ensures that even drones with limited processing capabilities can participate in the consensus process, promoting scalability and energy efficiency.

Serein [100] introduces a parallel pipeline-based directed acyclic graph (DAG) schema to improve the efficiency of consensus in blockchain systems. The parallel pipeline architecture ensures that multiple drone transactions can be processed concurrently, enabling real-time coordination between drones without delays caused by waiting for consensus from previous transactions.

The Petrichor protocol [101], same as Komorebi [102], employs a DAG structure within shards to leverage parallel transaction processing. It introduces a leader block election mechanism to facilitate fast consensus within each shard, significantly improving performance. The asynchronous nature and parallel processing capabilities of the Petrichor protocol can reduce communication delays, ensuring faster consensus in dynamic environments like drone swarms.

TidyBlock [103] focuses on organizing IoT data efficiently within a DAG-based blockchain, enhancing transaction collation and verification processes to maintain high throughput and improve data analysis efficiency.

##### Decentralized Task Allocation and Coordination

DAG-based blockchains enable decentralized task assignment within a swarm, ensuring that tasks are distributed evenly without the need for central coordination [104]. This allows drone swarms to dynamically adjust to changes in their environment, such as reallocating drones in response to task failures or environmental obstacles.

DAGmap [105], a novel approach for simultaneous localization and mapping (SLAM) in multi-drone systems, focuses on overcoming the limitations of centralized systems by using a DAG-based distributed ledger to enable decentralized map sharing and reconstruction. The proposed system allows multiple drones to share 3D map data continuously without relying on a central station, using DAG to organize map features and ensure data integrity.

Duan et al. [106] utilizes DAG task scheduling in computing power networks. It employs distributed agents to assess local network states and formulate scheduling policies, enhancing task allocation and resource management.

##### Microtransactions and Resource Sharing

Drones in a swarm may need to exchange resources, such as battery power or computational capacity, to optimize mission performance. DAG-based blockchains enable feeless microtransactions between drones, facilitating efficient resource sharing and promoting cooperative behavior within the swarm.

RT-DAG [107] facilitates rapid and concurrent transaction processing, enabling seamless microtransactions within a distributed network, which is crucial for resource allocation and services in drone swarms.

With the work of Zheng et al., a DAG-based blockchain architecture has been proposed for secure sharing of big data among multiple institutions [108]. This architecture includes a design for secure data sharing and efficient data management, which is crucial for resource sharing.

Jiang et al. [109] addresses the challenge of energy management in UAV networks by proposing a system that uses blockchain to incentivize resource cooperation, including wireless power transfer. DAG-based blockchain is introduced to reduce transaction costs and manage micropayments between UAVs for energy trading.

Table 9 summarizes the contributions of various works that integrate DAG-based blockchain into drone swarm systems. It presents a clear overview of how these studies leverage DAG’s advantages such as scalability, low latency, energy efficiency, decentralization, and feeless transactions. The *Leverage of DAG’s Advantages* column evaluates the extent to which these works highlight and utilize the five specific advantages of DAG-based blockchain. The ratings—ranging from “High” to “Limited”—are based on how much emphasis the study places on each characteristic in relation to drone swarm operations. For instance, studies that focus heavily on real-time communication and transaction validation, such as those utilizing Serein [100] or Petrichor [101], tend to score higher on Scalability and Low Latency, as these characteristics are central to the research. On the other hand, works that emphasize security and task allocation, like those of Carovilla et al. [104], are rated lower on aspects like Feeless Transaction and Energy Efficiency, as their main focus is not directly on transaction processing but on system security.

**Table 9 sensors-25-01108-t009:** Summary of related works —DAG-blockchain in drone swarm.

Reference	Main Contribution	Relevance to DAG Blockchain	Leverage of DAG’s Advantages				
			Sc. ^1^	L.L ^2^	E.E. ^3^	De. ^4^	F.T. ^5^
Vas. [96] et al.	Develops a combat drone swarm management system using Solana blockchain, emphasizing secure, reliable, and scalable coordination.	Demonstrates the suitability of DAG for enhancing drone communication security and data integrity in military applications.	High	Moderate	Moderate	High	Limited
Chen [99] et al.	Propose a robust and efficient consensus mechanism for large-scale drone swarms to improve reliability and reduce failure risks.	Utilizes DAG to achieve robust consensus suitable for decentralized drone networks.	High	Moderate	Moderate	High	Limited
Liu [100] et al.	Introduce Serein, a parallel pipeline-based DAG schema for consensus in blockchain, optimizing latency and scalability through functional node partitioning.	Leverages DAG to increase consensus efficiency and throughput in blockchain environments, reducing latency and supporting large-scale decentralized systems.	High	High	Moderate	High	Limited
Peng [101] et al.	Proposes Petrichor, a DAG-based consensus mechanism for asynchronous Byzantine fault tolerance (BFT).	Applies DAG to achieve secure and efficient coordination among nodes in asynchronous settings.	High	High	Moderate	High	Limited
Peng [102] et al.	Introduces Komorebi, a DAG-based Asynchronous BFT approach using sharding.	Combines sharding and DAG to enhance scalability in swarm drone operations.	High	Moderate	Moderate	High	Limited
Qu [103] et al.	Proposes TidyBlock, a consensus mechanism tailored for IoT using DAG-based blockchain.	Improves reliability and efficiency in IoT deployments through DAG consensus.	High	High	Moderate	High	Limited
Carovilla [104] et al.	Investigates blockchain-based security for semi-centralized robotic swarms.	Enhances security and coordination in semi-centralized systems through DAG-based blockchain.	Moderate	Moderate	Moderate	Moderate	Limited
Park [105] et al.	Develops DAGmap, a multi-drone SLAM system using DAG-based blockchain.	Utilizes DAG blockchain for secure mapping and localization among multiple drones.	High	High	Moderate	High	Limited
Duan [106] et al.	Develops a DAG-based scheduling mechanism for UAV computing networks to improve task efficiency.	Demonstrates DAG’s relevance for scheduling complex tasks across UAVs efficiently.	High	High	High	High	Limited
Zheng [108] et al.	Explores data management across institutions using DAG blockchain, focusing on secure and scalable data sharing.	Highlights the potential of DAG blockchain in handling large-scale data securely.	High	Moderate	High	High	Limited
Gao [97] et al.	Proposes a DAG-based architecture for edge service in drones with 5G integration to enhance security and efficiency.	Uses DAG for seamless service provision in automated drone systems.	High	Moderate	High	High	Limited
Alkhalifa [98] et al.	Uses a Bayesian approach combined with DAG blockchain to enhance vehicular network security.	Adopts DAG blockchain to secure vehicle-to-drone communication in vehicular applications.	High	High	Moderate	High	Limited
Jiang [109] et al.	Proposes a mechanism for incentivizing resource cooperation for UAV power transfer	Employs DAG blockchain for efficient energy distribution and resource sharing among UAV networks.	Moderate	High	High	High	High

^1^ Scalability. ^2^ Low Latency. ^3^ Energy Efficiency. ^4^ Decentralization. ^5^ Feeless Transaction.

### 5.2. Robotic Systems

#### 5.2.1. RQ2: Application Scenario

Robotics has become a key driver in advancing smart mobility, particularly with the rise of autonomous systems [110]. These systems—comprising autonomous vehicles (AVs) [111,112,113], delivery robots [6,114], and automated infrastructure maintenance [115] solutions—leverage cutting-edge robotics technologies to enhance efficiency, safety, and sustainability in urban environments. All of them require secure coordination in unpredictable environments. Centralized control systems introduce single points of failure, while traditional blockchains incur prohibitive latency for real-time operations like obstacle avoidance.

#### 5.2.2. RQ3: Technical Features Support

DAG-based blockchain technology is increasingly being integrated across various domains of robotic systems, providing several benefits in terms of security, scalability, and efficiency. Below are key areas where DAG-based blockchain significantly enhances robotic system operations:

##### Secure and Scalable Communication in Swarm Robotics

In robotic swarms, multiple robots must coordinate to achieve collective goals, such as search and rescue or environmental monitoring. DAG-based blockchains enable secure, decentralized communication, preventing bottlenecks associated with traditional blockchains, while ensuring real-time data exchange and decision-making.

Luo et al. [116] introduce a partition-tolerant distributed ledger protocol that supports swarm robotics. It leverages DAG to manage communication and coordination across a swarm of robots, even in the presence of network partitions. The system enhances fault tolerance, allowing robots to communicate securely and in a scalable manner. This approach is critical for enabling microtransactions and resource sharing among robots in decentralized environments like drone swarms.

##### Decentralized Task Allocation

DAG-based blockchain facilitates autonomous task allocation within robotic fleets, reducing the need for centralized control systems. This improves the system’s flexibility and robustness, allowing robots to adapt dynamically to changing environments and redistribute tasks in the event of failure. A notable application is in multi-drone Simultaneous Localization and Mapping (SLAM. Traditional block-based DLTs face challenges such as high power consumption and restricted data structures.

A DAG-based approach, named DAGmap [105], addresses these issues by enabling efficient online map convergence and data validation among multiple drones without a centralized station.

##### Fault Tolerance

Fault tolerance in robotic systems is crucial for ensuring reliable operation, particularly in hazardous or remote environments [117].

DEXON [118] is a DAG-based consensus protocol designed to offer high scalability and decentralization while maintaining Byzantine fault tolerance (BFT). Its core innovation is its block–lattice architecture, which allows multiple nodes to propose blocks concurrently, reducing latency and increasing throughput. The DAG structure helps resolve conflicts between blocks and ensures that the network can process many transactions in parallel, making it ideal for high-throughput applications such as decentralized systems of autonomous robots.

Hydra [119] is another DAG-based consensus protocol designed to achieve asynchronous BFT with low latency and high scalability. In robotic systems, Hydra’s consensus mechanism ensures that, even in the presence of unreliable nodes, the robots can still coordinate and achieve tasks. Its DAG structure provides flexibility in communication between robots, allowing for dynamic changes in the network topology (e.g., robots joining or leaving the network), making it highly suitable for swarm robotics applications. Additionally, the protocol’s efficient handling of faults and its ability to reach consensus asynchronously is critical for maintaining the reliability and security of robot interactions in real-time operations.

##### Microtransactions and Transaction Processing

Robots can engage in feeless microtransactions for services such as maintenance or access to resources. This capability supports the development of autonomous “robot economies”, where machines efficiently transact and collaborate in real time.

The SDVP model [120] uses a self-referencing DAG structure combined with a voting-based PBFT consensus algorithm, which significantly improves transaction speed and reliability. This model demonstrates high efficiency in transaction processing, making it suitable for real-time robotic applications.

Table 10 summarizes the contributions of various works that integrate DAG-based blockchain into robotic systems. Similar to the evaluation criteria used in Table 9, this table provides a clear overview of how different studies leverage DAG’s advantages, including scalability, low latency, energy efficiency, decentralization, and feeless transactions.

**Table 10 sensors-25-01108-t010:** Summary of related works—DAG-blockchain in robotic systems.

Reference	Main Contribution	Relevance to DAG Blockchain	Leverage of DAG’s Advantages				
			Sc. ^1^	L.L ^2^	E.E. ^3^	De. ^4^	F.T. ^5^
Luo [116] et al.	Introduces a partition-tolerant distributed ledger protocol for swarm robotics, focusing on enhancing fault tolerance and secure communication.	Utilizes DAG to support decentralized, scalable communication in swarm robotics.	High	High	Moderate	High	Limited
Chen [118] et al.	Proposes DEXON, a highly scalable DAG-based consensus algorithm designed for low latency and high decentralization. Introduces a sharding framework and a novel VRF-based Byzantine Agreement protocol to enhance scalability and reduce communication overhead.	Utilizes DAG for decentralized, fault-tolerant consensus in high-throughput environments.	High	High	High	High	High
An [119] et al.	Proposes Hydra, a DAG-based asynchronous BFT protocol designed to ensure low latency and scalability in distributed systems.	Utilizes DAG to provide efficient consensus in asynchronous settings.	High	High	Moderate	High	Limited
Cao [120] et al.	Introduces SDVP, a voting-based DAG consensus mechanism for real-time robotic applications, emphasizing reliability and transaction speed.	Uses a DAG structure to enhance transaction processing speed and reliability in robotic systems.	High	High	Moderate	High	High

^1^ Scalability. ^2^ Low Latency. ^3^ Energy Efficiency. ^4^ Decentralization. ^5^ Feeless Transaction.

### 5.3. Electric Vehicles: Grid-Integrated Mobility

#### 5.3.1. RQ2: Application Scenarios

EV ecosystems involve complex interactions between vehicles, charging stations, and smart grids. Key challenges include securing vehicle-to-grid (V2G) energy trades, managing terabyte-scale telemetry data, and preventing billing fraud in decentralized environments.

#### 5.3.2. RQ3: Technical Feature Support

DAG-based blockchain technology has emerged as a promising solution to address several limitations of traditional blockchain systems in EV applications. By facilitating parallel transaction processing, DAG-based blockchains offer substantial advantages in terms of scalability, low latency, and feeless microtransactions. These advantages are particularly important for the high-frequency data exchanges and decentralized decision-making required in the EV ecosystem.

The following section explores various research efforts that integrate DAG-based blockchain into EV systems, highlighting its key applications and benefits:

##### Secure and Scalable Communication

By leveraging the decentralized, parallel processing nature of DAG, EVs can securely exchange data and conduct transactions in real time, without the bottlenecks or security vulnerabilities of traditional blockchain systems. This technology is key to enabling efficient, scalable, and trustworthy communication in the growing smart mobility infrastructure, making it ideal for the dynamic and interconnected EV ecosystem.

Aldweesh et al. [121] introduce a data authentication algorithm that leverages DAG-based blockchain to support real-time and scalable communication between vehicles and infrastructure. By using DAG’s inherent parallel processing capability, multiple data transactions can occur simultaneously, significantly enhancing the communication bandwidth. One of the primary advantages of DAG-based blockchain is its ability to minimize transaction delays. In an IoV environment where real-time data transmission is critical for vehicle coordination and decision-making, the low-latency nature of DAG structures ensures that vehicles can share and process data with minimal delay.

Chai et al. [122] introduce a DAG-based blockchain framework that supports the scalable sharing of knowledge among intelligent connected vehicles. The parallel processing capabilities of DAG allow multiple vehicles to exchange data simultaneously without bottlenecks, significantly improving the network’s ability to handle high volumes of data in real time. This ensures that, even as the number of connected vehicles grows, the system maintains efficient communication.

##### Smart Contracts and Energy Trading

Smart contracts on DAG-based platforms can automate and secure energy trading processes. This is particularly useful for managing distributed energy resources (DERs) and integrating EVs into smart grids for energy storage and supply.

Florea et al. [123] propose a DAG-based blockchain system that supports the scalable management of battery data in electric vehicles. Unlike traditional blockchain, DAG-based blockchain allows for the parallel processing of multiple transactions, enabling EVs to share and process battery-related information in real time. This scalability is essential as the number of EVs grows, ensuring that the network can handle a high volume of data without compromising performance. This paper highlights how DAG-based blockchain facilitates feeless microtransactions, which are crucial for enabling energy trading between EVs and the grid. This supports the vehicle-to-grid (V2G) concept, where EVs can sell excess energy back to the grid or trade energy with other vehicles. The absence of transaction fees ensures that even small-scale energy trades are economically viable, improving the overall efficiency of the energy management system.

Halgamuge et al. [124] introduce the use of DAG-based blockchain, specifically IOTA’s Tangle, to facilitate the secure exchange of data between vehicles and infrastructure through smart contracts. This allows vehicles to autonomously execute contracts without needing centralized authority, ensuring that all agreements (such as data sharing or energy trading agreements) are immutable and tamper-proof. This paper emphasizes throughput optimization for energy and data transactions. DAG-based blockchain’s ability to handle transactions in parallel, unlike traditional blockchain, ensures that the network can support high volumes of transactions simultaneously.

Hassija et al. [125] leverages smart contracts on a DAG-based blockchain to automate energy trading transactions. Vehicles can autonomously buy or sell energy based on predefined rules, such as price thresholds or energy needs. These smart contracts ensure that energy transactions are executed securely and without intermediaries, reducing the cost and complexity of managing energy trades between vehicles and the grid. Meanwhile, the introduced framework supports feeless microtransactions, which is crucial in the context of V2G energy trading. Vehicles often trade small amounts of energy in frequent, low-cost transactions. Traditional blockchain systems may impose fees that make such microtransactions impractical. DAG-based systems, by contrast, allow these trades to occur without transaction fees, making small-scale, frequent energy exchanges economically viable.

##### Enhanced Security and Trust

The decentralized and immutable nature of DAG-based blockchains ensures that all data related to charging, energy usage, and V2G transactions are secure from tampering, creating trust in the EV ecosystem. This enhances the transparency of EV operations and reduces the risk of fraud.

Firooajaei [126] introduces a framework that leverages blockchain technology to enhance the transparency, trust, and security of electric vehicle (EV) charging networks. By using DAG-based blockchain, the system becomes highly resistant to attacks like double-spending and data manipulation. This paper highlights that the decentralized nature of the DAG framework ensures that malicious actors cannot easily alter past transactions, even if they control several nodes in the network. This is particularly beneficial in EV charging networks where trust between users and charging stations is paramount.

Das et al. [127] introduce a trust-based authentication scheme that ensures secure communication between vehicles and infrastructure in an IoV system. By using DAG-based blockchain, the system can authenticate multiple transactions in parallel, ensuring that each interaction is secure without the delays caused by traditional blockchain’s sequential block validation.

Aldweesh et al. [121] introduce a data authentication algorithm designed to ensure that all information shared across IoV systems is verified and tamper-proof. By leveraging DAG-based blockchain technology, the system ensures that data from vehicles and roadside units (RSUs) are immutably recorded, providing enhanced data integrity and preventing unauthorized alterations to critical information. The algorithm enhances security by mitigating Sybil attacks, where an attacker might create numerous fake identities to manipulate the network. The use of DAG-based blockchain ensures that any attempt to introduce multiple malicious nodes is prevented by its decentralized and distributed ledger, which verifies each node’s legitimacy before allowing it to participate in the network.

Duan et al. [128] propose a framework that incorporates a secure data-sharing mechanism using DAG-based blockchain, which ensures that sensitive information—such as energy usage data, transportation data, and public service management—can be shared securely and efficiently across different city departments and stakeholders. By employing DAG-based blockchain, the system allows for trustless interaction among various smart city services (e.g., transportation, energy management, waste management). Each service operates independently yet collaboratively within a trusted, decentralized environment, where data authenticity is guaranteed without requiring intermediary validation.

Table 11 provides a summary of the contributions from various works that integrate DAG-based blockchain into EV systems. Similar to the evaluation criteria used in the drone swarm table, this table outlines how these studies utilize DAG’s advantages, such as scalability, low latency, energy efficiency, decentralization, and feeless transactions. The table also includes an evaluation of the relevance of these characteristics in each study, providing insight into how DAG-based blockchain technology is applied to enhance the EV ecosystem.

**Table 11 sensors-25-01108-t011:** Summary of related works—DAG-blockchain in EV systems.

Reference	Main Contribution	Relevance to DAG Blockchain	Leverage of DAG’s Advantages				
			Sc. ^1^	L.L ^2^	E.E. ^3^	De. ^4^	F.T. ^5^
Aldweesh [121] et al.	Proposes a data authentication algorithm for real-time and scalable communication in vehicular networks using DAG blockchain.	Utilizes DAG for secure and parallel data sharing in smart mobility applications.	High	High	Moderate	High	Limited
Chai [122] et al.	Develops a DAG-based blockchain framework to support scalable knowledge sharing among connected vehicles.	Employs DAG for parallel data exchange between vehicles.	High	High	Moderate	High	Limited
Florea [123] et al.	Introduces a DAG-based blockchain system for scalable management of battery data in electric vehicles.	Leverages DAG to enable real-time data processing for energy management in EVs.	High	High	Moderate	High	High
Hal. [124] et al.	Proposes the use of DAG-based blockchain, specifically IOTA’s Tangle, for the secure exchange of data between vehicles and infrastructure.	Uses DAG to enhance the efficiency of smart contracts and secure data exchanges.	High	High	Moderate	High	Moderate
Hassija [125] et al.	Utilizes DAG-based blockchain for automated energy trading transactions between electric vehicles.	Applies DAG to enable secure, low-cost energy trades in EV networks.	High	High	Moderate	High	High
Fir. [126] et al.	Develops a framework leveraging DAG blockchain for enhanced transparency and security in EV charging networks.	Highlights DAG’s benefits for secure transaction handling in EV charging.	High	Moderate	Moderate	High	Limited
Das [127] et al.	Introduces a trust-based authentication scheme for secure communication in vehicular IoT systems using DAG blockchain.	Uses DAG to enhance real-time secure communications between vehicles and infrastructure.	High	High	Moderate	High	Limited
Duan [128] et al.	Proposes a secure data-sharing framework using DAG blockchain for smart city applications.	Demonstrates DAG’s potential in supporting cross-domain data integrity in smart city environments.	High	High	Moderate	High	Limited

^1^ Scalability. ^2^ Low Latency. ^3^ Energy Efficiency. ^4^ Decentralization. ^5^ Feeless Transaction.

## 6. Technical Barriers to Deploying DAG-Based Blockchains in Smart Mobility Systems

This section systematically addresses RQ4, as framed by the PRISMA-guided review: What are the major technical barriers (e.g., security, scalability, consistency) to deploying DAG-based blockchains in smart mobility systems? We analyze four interconnected challenges—consensus mechanism complexity, probabilistic finality, structural inconsistencies in transaction ordering, and security vulnerabilities—that hinder the seamless integration of DAG architectures into latency-sensitive and safety-critical smart mobility ecosystems. Each subsection delineates the technical root causes, operational implications, and unresolved research gaps.

### 6.1. Decentralized Consensus Mechanisms: Balancing Scalability and Reliability

DAG-based blockchains replace linear block sequencing with graph-structured transaction validation, eliminating miners and enabling parallel processing. While this paradigm enhances scalability, it introduces inherent complexities in achieving decentralized consensus—a critical requirement for trustless smart mobility networks.

#### 6.1.1. Trade-Offs in Tip Selection Algorithms

The absence of miners shifts validation responsibilities to network participants via probabilistic tip selection mechanisms (e.g., MCMC random walks in IOTA [129]). These lightweight algorithms reduce energy consumption but struggle to maintain efficiency as network density increases. In vehicular networks with fluctuating node participation, delayed tip resolution can propagate latency across decision-making pipelines, jeopardizing real-time collision avoidance or route optimization [102].

#### 6.1.2. Centralization Dependencies During Bootstrapping

Early-stage DAG networks often rely on centralized coordinators (e.g., IOTA’s Coordinator) to prevent double-spending attacks until sufficient decentralization is achieved [130]. This interim centralization contradicts the autonomous ethos of smart mobility systems, where centralized points of failure could disrupt vehicle-to-everything (V2X) communications or enable adversarial manipulation of traffic data flows.

#### 6.1.3. Challenges Related to the Complex Consensus Mechanism

In smart mobility, where real-time data processing, rapid decision-making, and secure communications are paramount, the complexity of the DAG-based consensus mechanism can have several impacts:

##### Latency in Decision-Making

The probabilistic nature of the consensus mechanism and the complexity of tip selection can introduce latency, especially in larger networks where nodes must evaluate multiple transactions before approving new ones. In autonomous vehicle networks or drone swarms, this could delay critical decisions, potentially compromising safety and operational efficiency.

##### Security Vulnerabilities

The complex consensus algorithms, while lighter than PoW, are still vulnerable to certain types of attacks, such as parasite chains [131] or lazy nodes, which attempt to game the system by attaching invalid transactions. In the context of smart mobility, where trust and security are crucial, this could lead to system failures or security breaches, undermining the integrity of the entire network.

##### Scalability Issues

As smart mobility systems scale, with potentially thousands of vehicles or drones interacting on a decentralized network, the computational load of managing consensus in a DAG-based system can increase. This can lead to congestion, where too many transactions are left unapproved, slowing down the overall network performance [132].

The complexity of the consensus mechanism in DAG-based blockchain systems, particularly in managing decentralized validation and ensuring security without a centralized authority, presents significant challenges in smart mobility applications. This complexity can lead to issues with latency, security vulnerabilities, and scalability, all of which are critical in ensuring the safe and efficient operation of autonomous vehicles and drone swarms. While DAG-based systems offer promising solutions for scalability and low-latency transactions, overcoming these challenges remains a key area of ongoing research.

### 6.2. Probabilistic Finality: Uncertainty in Transaction Irreversibility

In DAG-based blockchain systems, finality refers to the point at which transactions are considered irreversible and permanently valid within the network [16]. Unlike traditional blockchain systems that often rely on deterministic finality (where a transaction is immediately confirmed and cannot be reversed), DAG-based blockchains typically achieve probabilistic finality. This means that the likelihood of a transaction being reversed decreases over time as more transactions are added to the network. While this approach enhances scalability, it introduces specific challenges in applications such as smart mobility, where real-time, reliable decision-making is critical.

#### 6.2.1. Challenges Related to the Probabilistic Finality

##### Reliability and Trust in Autonomous Decision-Making

In smart mobility applications, such as autonomous vehicles (AVs) and drone swarms, the ability to make quick and trustworthy decisions based on data is paramount [133]. Probabilistic finality introduces uncertainty in this process because transactions (or data points) are not immediately guaranteed to be permanent. Instead, they reach finality over time as they are “buried” deeper in the DAG structure, reducing the risk of reversal [134]. This delay in achieving finality can create issues in systems like vehicle-to-vehicle (V2V) communication or traffic management, where decisions must be based on the latest data. For instance, if an autonomous vehicle is making a decision based on traffic data that are still in a probabilistic state of finality, there is a risk that the data may later be reversed or altered, potentially leading to unsafe or inefficient decisions.

##### Vulnerability to Attacks

The risk of a reversal in probabilistic finality makes DAG-based systems more susceptible to adversaries with significant computational power. An attacker could, theoretically, exploit this feature by reorganizing transactions, especially during periods when the system has not yet achieved strong consensus [135]. This poses a challenge in critical infrastructure such as smart mobility, where integrity is vital for safety. For example, an attacker could attempt to delay or reverse a transaction that communicates critical traffic updates between AVs or drone swarms. Such disruptions could lead to accidents, inefficient routing, or system malfunctions. The need to continuously ensure the security of probabilistic finality is a major challenge for DAG-based blockchain systems in smart mobility applications.

##### Latency and Performance

In smart mobility, real-time data exchange is crucial. Systems like AVs rely on ultra-low latency to make quick decisions [136]. However, the probabilistic nature of finality in DAG-based blockchains can introduce a delay before transactions are considered confirmed. While this delay is often negligible in smaller networks, it can become significant as the network grows or becomes congested [134]. In large-scale implementations such as smart cities, where numerous AVs and drones need to communicate, even a minor delay in transaction finality could result in performance bottlenecks or delays in critical decision-making processes.

##### Role of Trusted Roles (TRs)

Some DAG-based systems attempt to mitigate the risks of probabilistic finality by incorporating Trusted Roles (TRs) to achieve deterministic finality. These trusted nodes or entities help establish a form of consensus that locks transactions in place more quickly. For example, Byteball [63] relies on a set of reputable witnesses to determine the main chain with finality, and Nano [77] uses voted representatives to confirm transactions deterministically. However, the introduction of TRs partially reintroduces centralization, which can undermine the benefits of decentralization inherent to blockchain systems. In the context of smart mobility, incorporating TRs might speed up transaction finality but could also introduce new attack vectors, as the system now has specific nodes that, if compromised, could disrupt the network. Balancing the trade-offs between decentralized probabilistic finality and centralized deterministic finality remains an ongoing challenge for smart mobility applications.

The probabilistic finality inherent in DAG-based blockchain systems poses several challenges for smart mobility applications. While it allows for enhanced scalability and flexibility, the delayed certainty in transaction confirmation and the risk of reversal introduce vulnerabilities in systems that rely on real-time, reliable data, such as autonomous vehicles and drone swarms. These issues highlight the need for further research into mechanisms that can balance the trade-offs between achieving fast finality and maintaining decentralized security, ensuring that DAG-based blockchain systems can fully support the demands of smart mobility applications.

### 6.3. Consistency and Ordering in DAG Structure

In DAG-based blockchain systems, consistency and order are two critical issues that directly affect the reliability and functionality of applications such as smart mobility. Unlike traditional blockchains, where blocks are sequentially ordered and the consensus is relatively straightforward, DAG systems are structured more dynamically, which introduces challenges in maintaining strict consistency and reliable ordering of transactions [16]. These challenges become particularly evident in complex, real-time applications such as smart mobility, where autonomous vehicles, drones, and other connected devices require precise, timely, and consistent data.

Consistency in blockchain refers to the state where all nodes in the network have an agreed-upon and synchronized ledger. In traditional blockchain systems, such as Bitcoin, strict consistency is maintained because all nodes follow a single chain, ensuring that the latest state of the ledger is universally synchronized [137]. In contrast, DAG-based systems, which lack a single chain, often rely on trusted authorities or specific mechanisms (such as IOTA’s Coordinator) to achieve consistency. This reliance on external entities, or a subset of trusted nodes, to ensure that all transactions are valid and consistent across the network creates several complications [63,77].

In smart mobility applications, maintaining consistency is crucial, as any inconsistency in the data (e.g., traffic conditions, vehicle positions, sensor data) could lead to incorrect decisions by autonomous vehicles or drone swarms. For instance, if two vehicles have different views of the state of the network due to inconsistency in the blockchain, they may make conflicting decisions that could result in accidents. Thus, the reliance on external entities for consistency in DAG-based systems undermines the decentralized and autonomous nature of smart mobility, as these systems require a consistent and real-time global view to operate efficiently.

Ordering refers to the sequence in which transactions are arranged and validated within the blockchain [16]. In traditional blockchains, transactions are ordered linearly through blocks, and the sequence is universally agreed upon by all nodes. However, in DAG-based systems, transactions are attached to previous transactions in a non-linear manner, which leads to the formation of topological ordering rather than a strict linear sequence.

This poses a challenge for smart mobility applications, where real-time and sequential processing of transactions is essential. For example, in autonomous vehicle networks, the order in which traffic updates or vehicle positions are processed is critical for decision-making. Any delay or misordering of transactions can lead to incorrect assessments of the environment, jeopardizing safety. Moreover, DAG-based systems often require periodic sorting of transactions, which further complicates real-time decision-making. If vehicles or drones are waiting for transactions to be periodically sorted and confirmed, this latency could affect their ability to react promptly to changes in their environment.

#### 6.3.1. Challenges Related to Consistency and Ordering

The lack of strict consistency and linear ordering in DAG-based blockchain systems presents the following key challenges for smart mobility:

##### Inconsistent Data Across Nodes

Since DAG-based systems may not always provide strict consistency, different vehicles or drones in a smart mobility system could have conflicting views of the network’s state [138]. For example, one vehicle might believe a particular road is congested, while another does not, leading to conflicting routing decisions and inefficiencies.

##### Delayed Transaction Finality

In smart mobility, the finality of transactions needs to be achieved as quickly as possible. However, the reliance on topological ordering and probabilistic finality in DAG-based systems means that transactions may not be immediately recognized as final, leading to delays in processing critical data. This can negatively impact real-time applications like vehicle-to-vehicle communication or coordinated drone swarms, where every millisecond counts.

##### Difficulty in Developing Smart Contracts

One of the key innovations in blockchain technology is the use of smart contracts, which are self-executing contracts with the terms of the agreement directly written into code. For smart mobility applications, smart contracts could automate processes such as toll payments, vehicle servicing, or insurance settlements. However, in DAG-based blockchains, the lack of consistent transaction ordering complicates the execution of smart contracts [139,140]. Since smart contracts often rely on a clear and deterministic sequence of transactions, the non-linear structure of DAG makes it difficult to guarantee the correct execution order of contract conditions.

The challenges of achieving strict consistency and reliable ordering in DAG-based blockchain systems significantly impact their application in smart mobility. Inconsistencies across nodes and delays in transaction finality hinder the real-time decision-making processes required by autonomous vehicles and drones. Additionally, the non-linear transaction structure of DAG-based systems complicates the development of smart contracts, which rely on deterministic ordering. Overcoming these challenges is crucial for the successful implementation of DAG-based blockchains in smart mobility applications.

### 6.4. Security

Several mainstream attacks, such as parasite chain attacks, balance attacks, large weight attacks, censorship attacks, Sybil attacks, double spending attacks, and replay attacks, can significantly influence their application in smart mobility systems. In smart mobility, where real-time decision-making, data integrity, and reliability are paramount, these attacks can jeopardize the overall security and functionality of the system. Here IS how these security concerns impact smart mobility:

#### 6.4.1. Types of Attacks and Challenges

##### Parasite Chain Attack

This kind of attack [131] occurs when an attacker creates an alternate subgraph or “parasite chain” that competes with the honest graph by attaching invalid or conflicting transactions to the main DAG. The attacker generates conflicting transactions in secret and releases them when they have sufficient computational resources to outpace the honest participants. In smart mobility, such attacks could lead to inconsistent data about vehicle positions, traffic conditions, or coordination among autonomous systems. For instance, an autonomous vehicle (AV) may rely on outdated or manipulated data, leading to incorrect decisions that can result in accidents or inefficiencies. Popov et al. [141] implement a modified tip selection mechanism, first-order Markov Chain Monte Carlo (MCMC), to resist parasite chain attacks by making it increasingly difficult for an attacker to manipulate the DAG without holding significant computational resources.

##### Balance Attack

Balance attack [142] involves partitioning the network into several subgraphs that grow at the same rate, allowing an adversary to strategically move between them, profiting by forcing the system to prioritize their chosen transactions. In smart mobility, this could disrupt real-time transaction processing, such as toll payments, charging transactions, or data updates from traffic management systems. AVs may not receive accurate or timely information due to this attack, disrupting coordination and safety measures. Protocols such as GHAST [143,144], used in Conflux [143], counterbalance this attack by slowing the block generation rate and implementing a weighted block selection mechanism that minimizes the chances of such an attack succeeding.

##### Large Weight Attack

This kind of attack [135] occurs when an attacker builds a conflicting transaction with a large enough weight (e.g., cumulative approvals or confidence scores) to invalidate previously confirmed transactions. In a smart mobility network, this could allow attackers to invalidate legitimate transactions, such as vehicle coordination or toll payments. A transaction that a vehicle relies on for making decisions could be reversed or discarded, leading to system failures or unsafe operations. IOTA mitigates large-weight attacks by using first-order MCMC [141] to ensure that transactions near the tip of the DAG are more likely to be selected for validation, making it harder for an attacker to build a heavy conflicting chain. E-IOTA [145] can adjust the effectiveness of tip selection, providing a similar resistance mechanism.

##### Censorship Attack

In a censorship attack [146], adversaries prevent specific transactions from being included in the DAG by controlling enough validators or colluding with other malicious nodes to reject or delay certain transactions. In smart mobility, such an attack could prevent important data, such as vehicle-to-vehicle communication or sensor updates, from being processed. This can lead to stalled decision-making processes, reduced efficiency, and compromised safety, particularly in high-density urban environments where quick data flow is crucial. DAG-based systems like Prism [67] provide defenses against censorship attacks by ensuring that a fraction of nodes always behave honestly, even if a portion of the network is compromised. This approach reduces the probability of a successful censorship attack.

##### Sybil Attack

A Sybil attack [147] occurs when an adversary generates multiple fake identities or nodes to gain disproportionate control over the network. In a DAG-based blockchain, this can lead to manipulation of transaction validation and approval processes. In smart mobility, Sybil attacks can lead to control over critical nodes, allowing attackers to manipulate transaction outcomes, such as tampering with traffic updates, vehicle coordination, or even toll payments. This could cause severe disruptions, leading to gridlock or unsafe driving conditions. To defend against Sybil attacks, DAG-based blockchains like Blockclique [148] employ a Sybil-resistant selection mechanism that requires significant resource commitments from nodes attempting to join the network. This makes it costly for attackers to generate multiple fake identities.

##### Double Spending Attack

In this kind of attack [131], an adversary attempts to initiate double-spending between two branches of a DAG by identifying two subgraphs with nearly identical cumulative weights. The attacker then attaches conflicting transactions to these branches, effectively attempting to spend the same digital asset multiple times. By continuously sending meaningless transactions to ensure the growth of these branches at a similar rate, the adversary can maintain the conflicting status of the transactions over time. This strategy results in the long-term persistence of a fork, allowing the attacker to exploit the system for malicious activities or to gain profits during this interval. Unlike attacks that rely on miner participation, the double spending attack does not require explicit coordination with miners, which makes it simpler to execute. The attacker only needs to send large volumes of transactions to the target branches without engaging in complex actions that could raise suspicion or be detected by miners. In the context of smart mobility, a double spending attack could have severe consequences, especially in systems relying on accurate, real-time data for coordination among autonomous vehicles (AVs) or traffic management. For example, if two conflicting transaction branches contain manipulated data regarding vehicle locations or traffic conditions, it could lead to inconsistent or incorrect information being disseminated across the system. An autonomous vehicle could, therefore, rely on outdated or inaccurate data for navigation, leading to inefficient routing, unsafe driving decisions, or even accidents. Furthermore, such attacks could destabilize real-time systems managing traffic flow or EV charging stations, resulting in congested routes, energy shortages, or delayed services.

##### Replay Attack

A Replay attack [149] occurs when an adversary intercepts and replays a legitimate transaction multiple times in order to steal funds or manipulate system processes. This attack typically exploits the reuse of addresses in the transaction network. By reusing the same address, an attacker can trigger the re-execution of transactions that have already been confirmed, draining funds from any subsequent transactions linked to the same address. In the context of DAG-based smart mobility systems, a Replay attack could severely disrupt the integrity of the data being exchanged. If an attacker replays transactions related to vehicle location, traffic management, or autonomous vehicle coordination, it could lead to erroneous or outdated information being processed by the system. For instance, an autonomous vehicle (AV) may receive a replayed transaction indicating an incorrect vehicle position or traffic condition, causing it to make unsafe decisions, such as taking a wrong turn, failing to recognize a traffic signal, or even causing a collision. Additionally, such attacks could compromise the reliability of payment systems for EV charging or toll payments, where repeated, fraudulent transactions could lead to financial losses or disrupted services.

## 7. Future Research

This section systematically addresses RQ5 by examining critical challenges and proposing targeted research pathways to facilitate the broader adoption of DAG-based blockchain systems in smart mobility ecosystems. Our analysis identifies five interconnected research frontiers requiring concerted investigation.

### 7.1. Performance Evaluation

A notable challenge is the lack of standardized benchmarks for evaluating the performance of DAG-based consensus protocols. The absence of such benchmarks hinders objective comparisons and impedes the development of optimized systems. To address this, the development of comprehensive evaluation frameworks is essential. For instance, the DAGBENCH framework [150] allows for the measurement of DAG implementations in terms of throughput, latency, scalability, resource consumption, transaction data size, and transaction fees. Implementing such frameworks can provide valuable insights into the performance characteristics of DAG-based systems.

### 7.2. Security Analysis

Another significant challenge is the sporadic and varied nature of formal security proofs for DAG-based protocols. While some studies have provided formal specifications and proofs, they often differ in scope and depth. For example, a recent study presented a safety-proven formal specification of two DAG-based protocols, highlighting variations in dissemination, DAG construction, and ordering [151]. The TLA+ specification for a given protocol consisted of 492–732 lines, and the proof system TLAPS verified 2025–2294 obligations in 6–8 min. This indicates that, while formal verification is achievable, it requires substantial effort and may vary in complexity.

### 7.3. Scalability

Scalability remains a core challenge in DAG-based systems, particularly in high-transaction environments such as smart mobility. The selection of an appropriate tip selection algorithm is crucial for balancing performance and scalability. Future research should focus on optimizing consensus mechanisms and transaction throughput. Refining sharding techniques and developing hybrid consensus models that combine DAG with Byzantine fault tolerance (BFT) could address scalability limitations while maintaining high levels of security and decentralization. This would enable the processing of a higher volume of transactions with minimal delays, facilitating real-time applications like autonomous vehicle coordination and IoT-based services in smart cities. Further investigation is required to develop algorithms that effectively balance these factors.

### 7.4. Privacy and Robustness

Security is another critical concern that must be addressed to ensure the reliability of DAG-based blockchain systems. The distributed nature of these networks increases vulnerability to attacks, such as Sybil and balance attacks. Incorporating advanced cryptographic solutions, such as Zero-Knowledge Proofs (ZKPs) and secure multi-party computation (SMPC), could enhance data privacy and strengthen security. Furthermore, refining consensus protocols to ensure liveness and safety in the face of adversarial conditions would improve the robustness of DAG systems. Solving these security challenges would be essential for securing applications like toll payments, real-time vehicle-to-vehicle (V2V) communication, and vehicle-to-grid (V2G) energy trading. Further research is needed to develop robust security measures tailored to DAG-based systems.

### 7.5. Optimizing Feeless Microtransactions

DAG-based blockchain systems have a distinct advantage in enabling feeless microtransactions, which are crucial for use cases in smart mobility such as toll collection and ride-sharing. However, optimizing the transaction fee model to ensure that these transactions remain efficient and cost-effective without sacrificing security or scalability is still a challenge. Future research should explore off-chain scaling solutions and layer-2 protocols to improve transaction throughput and reduce computational costs. Achieving this would make DAG-based systems more viable for small-scale, high-frequency applications, paving the way for broader adoption in real-time services in the smart mobility sector. Additionally, the total ordering of transactions should be carefully considered to ensure fairness in transaction processing. Further investigation is required to develop mechanisms that maintain fairness while optimizing for feeless microtransactions.

## 8. Conclusions

This systematic review underscores the transformative potential of directed acyclic graph (DAG)-based blockchain technology in addressing critical challenges within smart mobility ecosystems. By synthesizing insights from 47 studies through PRISMA-guided analysis, the review demonstrates that DAG architectures significantly enhance scalability, latency, and energy efficiency compared to traditional blockchains. Key findings reveal that DAG-based systems achieve throughput exceeding 1000 TPS and sub-second latency, enabling real-time applications such as autonomous vehicle coordination, drone swarm operations, and EV charging microtransactions.

Despite these advantages, technical barriers persist, including consensus mechanism complexity, probabilistic finality, and vulnerabilities to attacks like double-spending and Sybil attacks. The decentralized nature of DAGs introduces challenges in transaction ordering and consistency, necessitating robust solutions for secure, large-scale deployments.

Future research priorities include:

Hybrid Consensus Models: Integrating DAG with Byzantine fault tolerance (BFT) to balance scalability and security.

Formal Security Frameworks: Developing proofs for DAG protocols to mitigate attack risks.

Standardized Benchmarks: Establishing metrics for performance evaluation across diverse smart mobility use cases.

Privacy-Preserving Techniques: Leveraging zero-knowledge proofs for secure data sharing.

Feeless Transaction Optimization: Enhancing microtransaction efficiency for IoT-driven mobility networks.

By addressing these challenges, DAG-based blockchains can catalyze the development of resilient, scalable smart mobility infrastructures, fostering sustainable urban transportation systems. This survey provides a foundational roadmap for researchers and practitioners to advance DAG innovations, ensuring their alignment with the dynamic demands of next-generation mobility ecosystems.

## Figures and Tables

**Figure 1 sensors-25-01108-f001:**
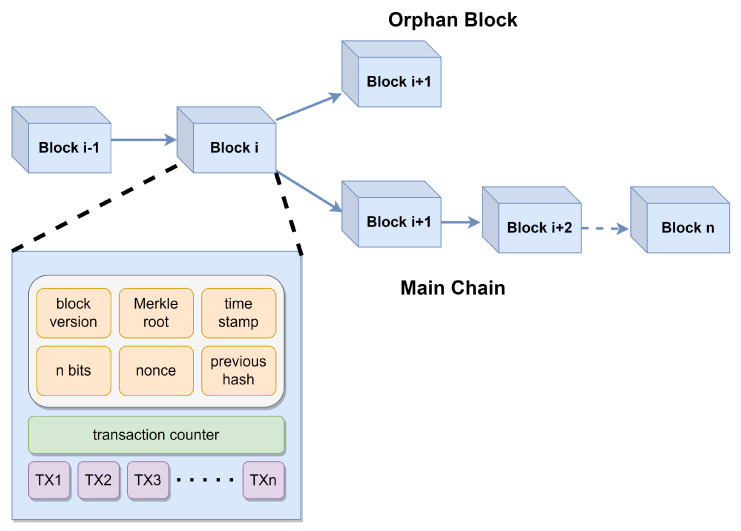
Structure of the sequential blockchain.

**Figure 2 sensors-25-01108-f002:**
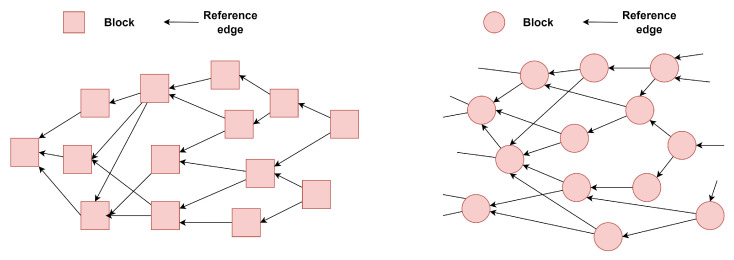
The structure of blocks and non-blocks based on the divergence topology. **Left**: Type II; **Right**: Type I.

**Figure 3 sensors-25-01108-f003:**
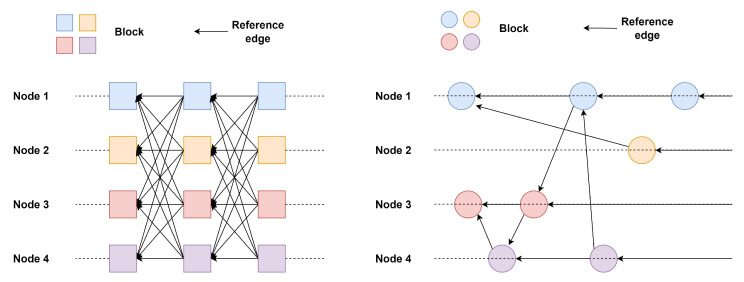
The structure of blocks and non-blocks based on the parallel topology. **Left**: Type IV; **Right**: Type III.

**Figure 4 sensors-25-01108-f004:**
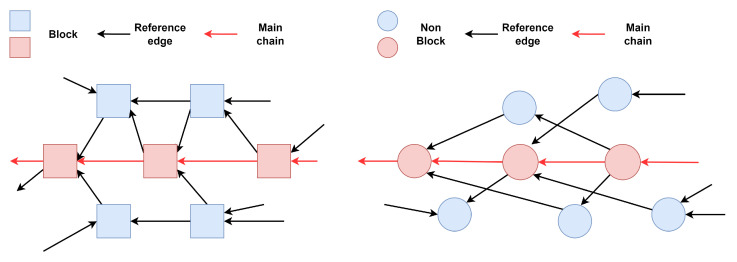
The structure of blocks and non-blocks based on the parallel topology. **Left**: Type VI; **Right**: Type V.

**Figure 5 sensors-25-01108-f005:**
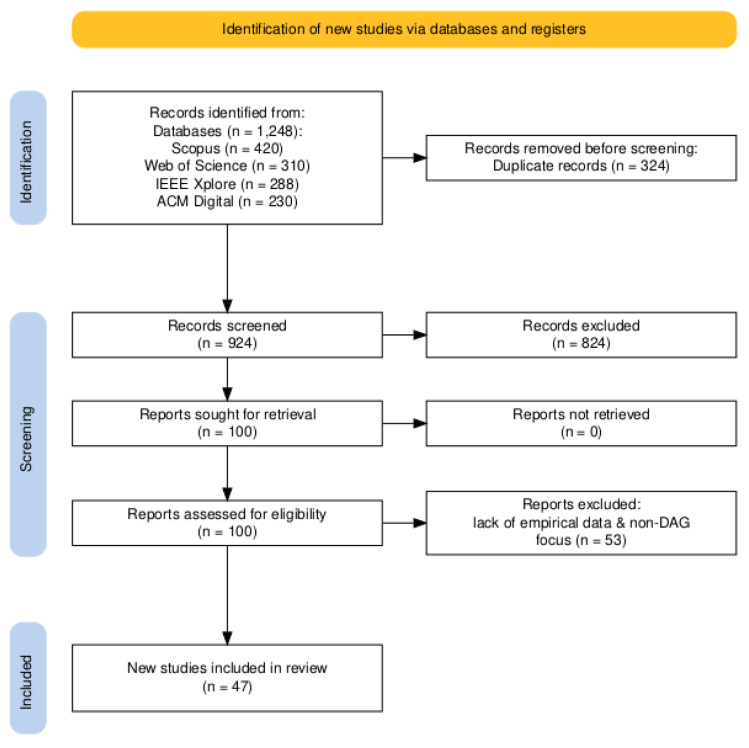
PRISMA flowchart [90].

**Figure 6 sensors-25-01108-f006:**
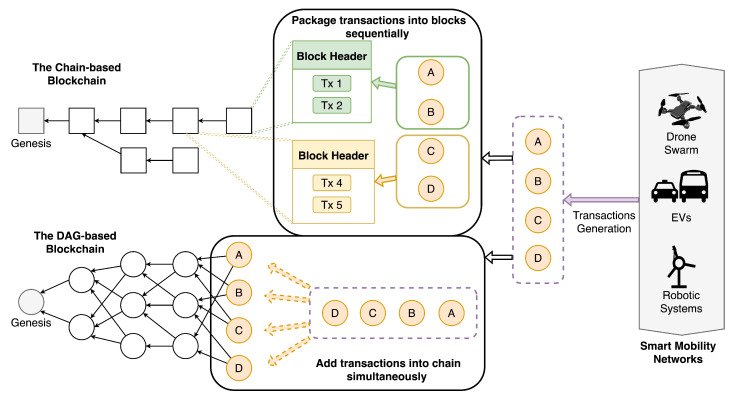
Chain-based blockchain and DAG-based blockchain in smart mobility.

**Table 1 sensors-25-01108-t001:** Summary of recent surveys on blockchain for smart mobility.

Ref.	Main Contribution	Applications			Desc. of DAG
		EVs	Robotics	Drone	
Srinivas Aditya et al. (2021) [17]	This paper explores the applications of blockchain in robotics, focusing on security, task distribution, and autonomous systems. Discusses how blockchain addresses critical issues in robotics like trust, data integrity, and fault tolerance.	-	✓	-	limited
Wu et al. (2021) [18]	This paper discusses blockchain-based solutions for 5G-enabled drone communications with a focus on privacy preservation. It reviews the architecture for 5G-enabled drone communications, privacy preservation using blockchain, and the existing legislation and challenges. It also explores privacy concerns, ID management, data privacy protection, trajectory privacy protection, and consensus in drone networks.	-	-	✓	limited
Kumar et al. (2022) [19]	This paper reviews the applications of blockchain in distributed control systems (DCS) and cooperative robotics, focusing on solving issues such as security and performance through decentralized frameworks.	-	✓	-	limited
Queralta et al. (2023) [20]	This paper reviews the integration of Distributed Ledger Technologies (DLTs), including blockchain and DAG-based systems, in decentralized multi-robot systems. It discusses the application of DLTs for secure, resilient, and scalable systems in robotic cooperation, coordination, and auditability. Key frameworks like Ethereum, Hyperledger Fabric, and IOTA are explored, highlighting their scalability and adaptability in real-world robotic systems.	-	✓	-	limited
Aung et al. (2020) [21]	This paper reviews blockchain applications in IoV (Internet of Vehicles) for security, privacy, trust management, and incentive systems. Highlights various blockchain frameworks and their use for secure communications, privacy protection, and traffic-related cryptocurrencies.	✓	-	✓	limited
Badidi (2022) [22]	This paper reviews the integration of Edge AI and Blockchain technologies for smart city applications, focusing on smart mobility and smart energy. It outlines the potential for these technologies to improve transportation, energy management, and overall urban sustainability, while discussing related research and future challenges.	✓	✓	✓	limited
Ajakwe et al. (2023) [23]	This paper systematically reviews security dynamics in Drone Transportation Systems (DTSs), focusing on AI, cybersecurity, airspace security, and blockchain technology. The authors emphasize the integration of blockchain for decentralizing authentication, authorization, and accountability in drone operations.	-	✓	✓	limited
Nour et al. (2022) [24]	This work provides a comprehensive review of blockchain applications in the electricity sector, discussing various barriers to adoption, such as high energy consumption, computational challenges, and scalability issues. Identifies future research areas including interoperability and the need for blockchain solutions tailored to the electricity sector.	✓	✓	-	limited
Our Paper	Our work provides a comprehensive overview of smart mobility, highlighting the integration of electric vehicles, robotic systems, and drone swarms, and addressing the challenges and opportunities of implementing DAG-based blockchain technology to enhance scalability, security, and efficiency in smart mobility systems.	✓	✓	✓	fully

**Table 2 sensors-25-01108-t002:** Comparison among public, private, and consortium blockchains [42,43].

Property	Public Blockchain	Private Blockchain	Consortium Blockchain
Consensus mechanism	All nodes	Single organization	Selected nodes
Immutability	Extremely high	Modifiable	Modifiable
Efficiency	Low	High	High
Centralization	Fully decentralized	Centralized	Partially centralized
Access permission	Permissionless	Permissioned	Permissioned
Participant identity	Pseudoymous	Known entities	Known entities
Transaction speed	Minutes	Milliseconds	Milliseconds
Examples	Ethereum [44]	Hyperledger Fabric [45]	R3 Corda [46]

**Table 3 sensors-25-01108-t003:** Comparison of different consensus protocols for sequential blockchain [47].

	Consensus	Throughput	Latency	Adversary Tolerance	Overhead		
					Computing	Network	Storage
Permissionless	PoW [48]	low	high	25% computing power	high	low	high
	PoC [49]	low	high	-	low	low	extremely high
	PoS [50]	low	medium	50% stakes	medium	low	high
	DPoS [51]	high	medium	51% validators	medium	-	high
	PoA [52]	low	medium	51% online stakes	high	low	high
	SCP [53]	high	medium	variable	low	medium	high
	Ripple [54]	high	medium	20% faulty UNL nodes	low	medium	high
Permissioned	PBFT [55]	high	low	33% faulty replicas	low	high	high
	dBFT [56]	high	medium	33% faulty replicas	low	high	high
	Tendermint [57]	high	low	33% voting power	low	high	high
	PoET [58]	high	low	-	low	low	high
	Raft [59]	high	low	50% crash fault	low	-	high

**Table 4 sensors-25-01108-t004:** Types of DAG-based blockchain system.

		Graph Topology		
		Divergence	Parallel	Convergence
Unit Representation	block-less	Type I	Type III	Type V
	block-based	Type II	Type IV	Type VI

**Table 5 sensors-25-01108-t005:** Summary of current DAG-based blockchain systems.

Openness	Consensus	Types	Writing Access ^1^	Leader-Based	Ordering	Finality
Permissionless	GHOST [61]	VI	PoW	Yes	Total	Probabilistic
	Inclusive [62]	VI	PoW	Yes	Total	Probabilistic
	Byteball [63]	V	Open	No	Total	Deterministic
	SPECTRE [64]	II	PoW	No	Partial	Probabilistic
	PHANTOM [65]	II	PoW	No	Total	Probabilistic
	DAG KNIGHT [66]	VI	PoW	No	Total	Probabilistic
	Prism [67]	IV	PoW	Yes	Total	Probabilistic
	Avalance [68]	I	Open	No	Partial	Probabilistic
	OHIE [69]	IV	PoW	No	Total	Probabilistic
	Chainweb [70]	III	PoW	No	Partial	Probabilistic
	Grahpchain [71]	I	PoW	No	Partial	Probabilistic
	Meshcash [71]	II	PoW	No	Total	Deterministic
	IOTA [72]	I	Open	No	Partial	Deterministic
Permissioned	Hashgraph [73]	III	Permissioned	No	Total	Deterministic
	Jointgraph [74]	III	Permissioned	Yes	Total	Deterministic
	Aleph [75]	III	Permissioned	No	Total	Deterministic

^1^ Control over the writing access defines who can contribute new information to the network: 1. PoW/PoS lottery based; 2. Permissioned: a pre-defined set of validators could write new blocks; 3. Committee: a rotating subset of validators could write new blocks; 4. Open writing access.

**Table 6 sensors-25-01108-t006:** Search string, databases consulted and the equation applied.

**Database**	Web of Science (WoS), Scopus, IEEE Xplore, and ACM Digital Library
**Cadena**	(“DAG blockchain” OR “IOTA” OR “smart mobility” OR “electric vehicles” OR “drone swarms” OR “robotic systems”) AND (“smart contract” OR “security” OR “consensus mechanism” OR “tip selection” OR “scalability”)

**Table 7 sensors-25-01108-t007:** Inclusion and exclusion criteria.

Criteria	Inclusion	Exclusion
Publication Type	Journal articles, conference proceedings, technical reports	Books, editorials, non-peer-review works
Language	English	Non-English publications
Focus	DAG-based blockchain applications in smart mobility subsystems	General blockchain studies unrelated to DAG or mobility
Technical Relevance	Empirical evaluations, case studies, theoretical frameworks	Opinion pieces, superficial reviews

**Table 8 sensors-25-01108-t008:** Advantages of DAG-based blockchain in smart mobility.

Advantage	Drone Swarm	Robotic Systems	Electric Vehicles (EVs)
Scalability	Enables real-time communication and coordination among large numbers of drones.	Facilitates decentralized task allocation and real-time data processing.	Supports the simultaneous processing of numerous transactions, including charging and V2G interactions.
Low Latency	Ensures quick decision-making for operations like emergency response.	Supports real-time communication for tasks like route optimization and sensor data processing.	Allows real-time interaction with charging stations and smart grids for optimized energy use.
Energy Efficiency	Uses lightweight consensus mechanisms, reducing energy consumption in drones.	Minimizes energy use while enabling decentralized decision-making in autonomous fleets.	Reduces energy consumption for EV networks by avoiding the computational overhead of traditional blockchains.
Decentralization	Allows autonomous decision-making and coordination without central authority.	Ensures each robot can operate independently while collaborating within the system.	Enables peer-to-peer interactions between EVs, charging stations, and grid operators.
Feeless Transactions	Supports feeless microtransactions for resource sharing within the swarm.	Facilitates efficient exchange of services without transaction fees.	Enables cost-effective energy trading, toll payments, and smart charging with feeless microtransactions.

## Data Availability

No new data were created or analyzed in this study. Data sharing is not applicable to this article.

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
