# Peer review of "A Survey on Directed Acyclic Graph-Based Blockchain in Smart Mobility"

_sensors, 2025, doi:10.3390/s25041108_

Round 1

Reviewer 1 Report

Comments and Suggestions for Authors

This paper has surveyed on DAG-based blockchain in the context of smart mobility. Graphical representation of the organization of the paper is good. Main technical challenges in Smart Mobility are highlighted well in Section 2 with solution of DAG-Based Blockchain as a New Trend in Smart Mobility in support with Figure 2 and details study of the DAG-Based Blockchain is done in Section 3. Section 4 has highlighted possible use cases with support of past literature in this domain. Challenges are highlighted in Section 5. Section 5.4 is very good as per obligatory in this type of study.

Now elaborate the future work more in Conclusion section with at-least a paragraph on it with highlight of how more efficiency, security and scalability can be gained with balances of these.

Reviewer 2 Report

Comments and Suggestions for Authors

The paper presents a review of DAG-based blockchain in smart mobility. The paper has sufficient contribution as it discusses DAG-blockchain in a complex scenario. Section 5 in the paper presents major challenges that guides researcher for further research. However, some minor modifications are needed

- Removing the abbreviation DAG from the title.

- section 1.1, the main verbs should be: studying, discussing, investigating ..

- Figure 1 section II is repeated , the green box should be section III

- adding spaces between reference number and the word 

- Since this is a review paper, it is important to distinguish between the authors findings and previous research finding. So, adding references to section 2.2 to prove these challenges, also adding ref to section 3.1, 3.2, 3.4.1, 3.4.2, 3.4.3. Also, ensure that the statements in section 5 are referenced.

- The structure can be improved as it is not clear why section 2.3 comes before section 3. there is overlap of ideas which need better arrangement 

- figure 2, adding more text to the caption to make the figure self-explained 

- Minor English check

Reviewer 3 Report

Comments and Suggestions for Authors

General comments

This paper presents a survey on the DAG-BC architectures and their potential application in the context of smart mobility.

Overall the domain of DAG-BC systems is interesting and addresses a series of the deficiencies of the early-primitive sequential BC systems.

The paper highlights some important attributes of the BC-based systems and the DAG-BCs such as the probabilistic/deterministic finality.

To the best of my efforts though, I found the paper to be an extensive literature review with self-contribution based on their own empirical data, no modeling or experimentation with the technologies/platforms of discourse, and no application-specific platforms’ suggestions.

To the best I can tell, the article will fall into the interest of readers interested in BC applications for the IoT world, as well as of readers interested in BC-based smart-mobility applications.

I suggest the paper to be accepted

Detailed comments
To the best of my understanding, the authors evaluation of the architectures covered in tables 3, 4 and 5 however useful to the reader, presents poorly supported estimations (High, Moderate, Limited) on their relevance to the targeted characteristics. No evidence of hands-on experience, experiments or evaluation methodology is presented.

Minor language and formatting issues throughout the paper, like in Page 10: typo “dorn” instead of “drone”. Paragraphs are often too lengthy, making it easy for the reader to lose the flow as in section 4.3. I suggest the authors to go through their paper to eliminate the minor language issues.

Overall, I suggest the paper is accepted for publication after amendments.

Reviewer 4 Report

Comments and Suggestions for Authors

This paper surveys the role of Directed Acyclic Graph (DAG)-based blockchain systems in smart mobility applications. It highlights the advantages of DAG technology, such as scalability, low latency, and fee-less micro-transactions, making it suitable for real-time, high-frequency transactions in smart mobility. The paper provides an extensive review of DAG-based blockchain technology and its application in diverse smart mobility scenarios like EVs, drone swarms, and robotic syste. The paper addresses a relevant and emerging topic but there are certain issues which need attention.

The paper contains excessive verbosity, making it difficult to follow at times. It is recommended that the author incorporate more illustrations and visual aids to enhance clarity and effectively communicate the information.

There are too many bullet points in paper and flow of the paper is not convincing in general. Sometimes it feels the information is presented in an orthogonal way. The writing lacks coherence, with some arguments requiring clearer articulation and connection to the research objectives.

Define smart mobility, along with its key features and goals, in the introduction. This addition will provide readers with a clear understanding of the concept and help them follow the paper more effectively.

It is unclear what specific research questions are being addressed in this study. I recommend that the author explicitly formulate and clearly articulate the research questions to provide a stronger focus and direction for the research.

What specific gap in the field is this paper aiming to address?

The paper lacks detailed information regarding the data collection process and the research methodology employed. I will suggest including a section discussing these details.

The review does it include a section on how different this review is from the previous ones published around smart mobilities

In several parts of the manuscript, the discussion lacks adequate support from references. It is strongly recommended to include appropriate citations to strengthen the arguments and provide credibility to the claims in the revised version. Provide references unless the information is general or not derived from the analysis results of the study.

Tables are presented in the manuscript; however, they lack adequate accompanying discussion, making it challenging for the reader to interpret and understand the information effectively. It is recommended to provide a detailed explanation and context for each table in the revised version.

Paper lacks technical depth in many sections. For example, in section 5.4.1 only 5 types of attacks are discussed with providing the information why other attacks are not discussed or not relevant.

Paragraphs 1 and 2 are too small on first page, merge them.

Revise the caption of Figure 1.

Equations on page 6 are not explained properly.

On page 7 line 263, the sentence is unclear

What is LL EE De in Table 3 and 4.

 By addressing these weak points, the paper could provide a more comprehensive, balanced, and impact-ful contribution to the field.

Round 2

Reviewer 4 Report

Comments and Suggestions for Authors

The revised version shows substantial improvements particularly in the methodology, results, and discussion sections. While the revisions improve the overall clarity, but article still has following shortcomings.

Figure 1 does not add any value, you may consider removing it.

I will suggest to provide a comparison of DAG-based blockchain with that traditional or sequential blockchain. I will recommend to provide a more comprehensive background on the field by defining key terms and concepts, especially for multidisciplinary audience.

Provide the exclusion criteria for the screening of articles following PRISMA approach. Consider revising Figure 2.

Although the authors has provided the the contribution but they have not clarified that what research questions are being addressed.

State limitations of the study if there are any.

I will recommend to improve the structure of the paper. Use subheading instead of too many bullet points.

The structure of the paper predominantly consists of short, disjointed paragraphs that lack a cohesive narrative or logical connection between ideas. This fragmented approach makes it challenging for the reader to follow the argument or understand the significance of the research. To improve readability and clarity, the authors are advised to re-organize the content into well-structured, logically sequenced paragraphs.

Figure 6 caption is too long.

Explain Table 4 and also cite it in the text. Ensure that every major argument is backed by a credible reference.

Due to the these issues, the manuscript cannot be accepted in its current form as it requires significant improvements in terms of structure, organization, and presentation. The current version lacks a cohesive flow, which makes it difficult for readers to follow the argument and fully grasp the contributions of the study. The structure needs to be revised to ensure that each section builds logically on the previous one, creating a seamless narrative from the research problem to the conclusions.
